# Conservative finite-volume forms of the Saint-Venant equations for hydrology and urban drainage

Ben R. Hodges[1]

[1]National Center for Infrastructure Modeling and Management, University of Texas at Austin

**Correspondence:** hodges@utexas.edu

**Abstract.** New integral, finite-volume forms of the Saint-Venant equations for one-dimensional (1D) open-channel flow are derived. The new equations are in the flux-gradient conservation form and transfer portions of both the hydrostatic pressure force and the gravitational force from the source term to the conservative flux term. This approach prevents irregular channel topography from creating an inherently non-smooth source term for momentum. The derivation introduces an analytical approximation of the free surface across a finite-volume element (e.g., linear, parabolic) with a weighting function for quadrature with bottom topography. This new free-surface/topography approach provides a single term that approximates the integrated piezometric pressure over a control volume that can be split between the source and the conservative flux terms without introducing new variables within the discretization. The resulting conservative finite-volume equations are written entirely in terms of flow rates, cross-sectional areas and water surface elevations – *without* using the bottom slope ($S_0$). The new Saint-Venant equation form is: (1) inherently conservative, as compared to non-conservative finite-difference forms, and (2) inherently "well-balanced" for irregular topography, as compared to conservative finite-volume forms using the Cunge-Liggett approach that rely on two integrations of topography. It is likely that this new equation form will be more tractable for large-scale simulations of river networks and urban drainage systems with highly-variable topography as it ensures the inhomogeneous source term of the momentum conservation equation is Lipschitz smooth as long as the solution variables are smooth.

## 1 Introduction

The one-dimensional Saint-Venant equations (SVE) are the simplest equations that capture the full dynamics of river and open-channel flow, and yet they are not universally used where river networks are explicitly represented in hydrological and urban drainage models. Furthermore, where the SVE are used the numerical solution methods typically employ a "non-conservative" form, which inherently has greater numerical dissipation than the "conservative" form. But even the undesirable non-conservative form of the SVE is little-used for large systems – instead, reduced-physics models are common. These approaches *a priori* neglect part some part of the flow dynamics for simpler computation (see Sections 2 and 3 for discussions of these issues). In their call for "hyperresolution global land surface modeling," Wood et al. (2011) postulated that the principle

issues that limit SVE use in river networks could be resolved by application of more computing power and more precise river geometry. Hodges (2013) argued that there were wider challenges to using the SVE, but these are not sufficient reasons relying on reduced-physics models that are calibrated to get the "right answer for the wrong reason." In the intervening years some progress has been made toward improving applications of the SVE, but large-scale hydrological modeling of river networks with application of supercomputer power continues to rely on reduced-physics methods (e.g., the US National Water Model, Cohen et al., 2018). Arguably, the dynamics of river flow are the most observable and should be the easiest part of hydrology to model (or at least with lower uncertainty in boundaries and forcing compared to overland or groundwater flow), so it is unsatisfactory that more than a half century after Preissmann (1961) we still are not universally using the SVE in hydrological modeling of river networks.

The present work evolved out of a frustration with the slow pace of improvement in SVE modeling. Taking a step backwards, we can ask – *is there something fundamental in the common forms of the SVE that hinders progress?* Motivated by an analysis of the equation forms (Section 2) and a study of the wealth of past work in the SVE (Section 3), new insights were developed and are presented herein. The fundamental theses of the present work are (1) conservative formulation of equations should be used for the next generation of river network models, and (2) the appearance of the channel slope ($S_0$) in the SVE for channels with irregular topography is a principle cause of instabilities and extended computational time. Neither thesis can be demonstrated herein – this work is merely a first step that provides the theoretical foundations for a conservative and inherently "well-balanced" approach that highlights the minimal level of approximations needed for an SVE form with irregular topography. It remains for future studies to compare models built on these foundations to the existing approaches to determine if the new forms provide significant numerical advantages.

The new conservative form of the SVE is developed with a goal of addressing challenges associated with modeling large-scale 1D flow network systems. In the process of developing the new form, we will encounter a philosophical question as to whether the primary vertical variable in a large-scale network solution should be the depth ($H$) or the water surface elevation ($\eta$). Despite this author's prior work with $H$ primacy (Liu and Hodges, 2014), we shall see that there are advantages to using $\eta$, which is identical to the piezometric pressure and hence uniform over a channel cross-sectional area. The quadrature of the subgrid piezometric pressure gradient and subgrid-scale topography can be handled in a single new term that is derived herein. Through this term arise interesting possibilities for analytically including hyperresolution bathymetric knowledge while retaining larger computational elements for large-scale modeling. This idea is not fully exploited within the present work, but the framework is developed for others to build upon.

In the remainder of this paper, Section 2 provides motivation and context in the differential forms of the SVE. Section 3 provides a further overview of SVE modeling in the wide range of conservative and non-conservative forms. A new and complete derivation of the finite-volume form of the conservative 1D momentum equation with minimal approximations is provided in Section 4. Approximate forms of the 1D SVE are presented in Section 5 and the final form of the equations and a discussion of their potential use is provided Section 6.

## 2 Motivation

In reach-scale hydraulic studies, the Saint-Venant equations are almost always solved in a conservative form (e.g., Karelsky et al., 2000; Lai et al., 2002; Papanicolaou et al., 2004; Sanders, 2001), but usually in non-conservative form when used in river-network hydrology and urban drainage networks (e.g., Liu and Hodges, 2014; Pramanik et al., 2010; Rossman, 2017; Saleh et al., 2013). Arguably, the reasons are in the difficulty in obtaining a "well-balanced" model for the conservative form and the inherent complexities/uncertainties of channel geometry across a large network (see discussion in Section 3). In general, conservative equation forms are valued as they ensure (with careful discretization) that transport modeling does not numerically create or destroy the transported variable. Indeed, the use of the conservative form for mass conservation is universal in models from hydraulics to hydrology – it is only for momentum that the non-conservative form remains common.

To set the context for this paper, consider the non-conservative form of the momentum equation that has been used in large river network solutions (Liu and Hodges, 2014)

$$\frac{\partial Q}{dt} + \frac{\partial}{\partial x}\left(\beta \frac{Q^2}{A}\right) + gA\frac{\partial H}{\partial x} = gA\left(S_0 - S_f\right) \tag{1}$$

where $Q$ is the flow rate, $A$ is the channel cross-sectional area, $\beta$ is the momentum coefficient (associated with non-uniform velocities integrated over $A$), $g$ is gravity, $H$ is the water depth, $S_0$ is the channel slope, and $S_f$ is the friction slope. The above equation has immediate physical appeal as each term represents a clearly understood piece of physics; i.e., the rate of change of the flow is affected by the gradient of nonlinear advection, the hydrostatic pressure gradient (driven by water depth), the gravitational force along the slope, and frictional resistance. The equation includes a conservative form of nonlinear advection (all the variables inside the gradient), but $gA\partial H/\partial x$ is non-conservative (that is $A$, which is a function of $H$, is outside the gradient). The terms to the right-hand-side (RHS) of the equal sign are considered source/sink terms that reflect creation and destruction of momentum. Thus, a "conservative" form of the above could be formally written as

$$\frac{\partial Q}{dt} + \frac{\partial}{\partial x}\left(\beta \frac{Q^2}{A}\right) = -gA\frac{\partial H}{\partial x} + gA\left(S_0 - S_f\right) \tag{2}$$

where the trivial act of moving the pressure gradient term to the RHS is a recognition that it can cause non-conservation (source/sink) of the conserved momentum fluxes from the left-hand-side (LHS). That is, any component on the RHS is capable of creating or destroying momentum whereas the terms on the LHS (if properly discretized) cannot. The above equations can be contrasted with an equivalent momentum equation using the free-surface elevation:

$$\frac{\partial Q}{dt} + \frac{\partial}{\partial x}\left(\beta \frac{Q^2}{A}\right) = -gA\frac{\partial \eta}{\partial x} - gAS_f \tag{3}$$

which is obtained by substitution of $\eta = H + z_b$ with $z_b$ as the bottom elevation and $S_0 = -\partial z_b/\partial x$. The equation is again "conservative" by virtue of including the gradient of the free surface elevation as a source term (Ying et al., 2004; Wu and Wang, 2007; Ying and Wang, 2008). Comparison of eqs. (2) and (3) shows how the introduction of $S_0$ effects the equation form. Of particular note is that we expect $S_f = f(Q, A)$, so the latter equation will have a smooth source term as long as the solution variables themselves are smooth. In contrast, the smoothness of the source term in eq. (2) inherently depends on the

smoothness of the product $AS_0$ and compensation by the solution in $A\partial H/\partial x$ and $AS_f$. The key point is that smoothness in the source term is a mathematical necessity for the numerical solution of a partial differential equation to be well-posed (Iserles, 1996), but introduction of $S_0$ can place smoothness at the mercy of how well the numerical scheme responds to non-smooth
forcing.

In general, there is an advantage to having as much of the physics as possible included on the flux-conservative side of the equation, which helps reduce difficulties associated with discretizations of the source terms (e.g., Pu et al., 2012; Vazquez-Cendon, 1999). The standard form of the conservative SVE is arguably the form provided by Cunge et al. (1980), based on a derivation of Liggett (1975):

$$\frac{\partial Q}{dt} + \frac{\partial}{\partial x}\left(\beta\frac{Q^2}{A} + gI_1\right) = gI_2 + gA\left(S_0 - S_f\right) \tag{4}$$

where $I_1$ and $I_2$ are integrated hydrostatic pressure forces across the channel

$$I_1 = \int_H (H - z)\,B\,dz \tag{5}$$

and along the channel gradients

$$I_2 = \int_H (H - z)\frac{\partial B}{\partial x}dz \tag{6}$$

with $H(x)$ as the water depth, $B(x,z)$ as the channel breadth as a function of elevation and along-stream location, and $z$ as a coordinate direction measured from a common horizontal baseline in a direction opposite to gravitational acceleration. This form is also used by others with slightly different nomenclature but the same integral terms (e.g. Hernandez-Duenas and Beljadid, 2016; Sanders, 2001; Saavedra et al., 2003). It will be convenient herein to call this the *Cunge-Liggett* form of the SVE. The key point for both these terms is that they measure the interaction between the free surface and the channel shape;
e.g. $I_1$ could also be written as

$$I_1 = \int_{z_b}^{\eta} (\eta - z)\,B\,dz \tag{7}$$

where $z_b$ is the channel bottom.

By comparing eq. (4) with eq. (2), we see that the novelty of Cunge-Liggett is in moving a portion of the $gA\partial H/\partial x$ from the RHS source term into the conservative flux on the LHS. Indeed, with this idea we see that eq. (2) can be used with the
product rule of differentiation to generate a slightly different conservative form:

$$\frac{\partial Q}{dt} + \frac{\partial}{\partial x}\left(\beta\frac{Q^2}{A} + gAH\right) = gH\frac{\partial A}{\partial x} + gA\left(S_0 - S_f\right) \tag{8}$$

Similar to the Cunge-Liggett form, the above uses a mathematical trick to place one part of the hydrostatic pressure force within the conservative flux gradient, while retaining the remainder as a source term. Thus, we see that the Cunge-Liggett form

is not canonical, nor is it a form that necessarily better represents the physics. It is merely a form that is (sometimes) convenient for splitting the gradient of the total hydrostatic pressure force into conservative flux and non-conservative source terms.

The above brings up a question: if it is good to shift a portion of the hydrostatic pressure from the source to the conservative term as in eqs. (4) and (8), then why not some or all of the gravitational potential associated with $gAS_0$? Underlying the Cunge-Liggett form and much (but not all) of the literature is the idea that $gAS_0$ is a source term that creates/destroys momentum. But this is also true of the hydrostatic pressure gradient and yet we commonly treat a portion within the convective flux. Thus, if Cunge-Liggett eq. (4) is preferred over the baseline conservative form of eq. (2) because a portion of the hydrostatic pressure is moved from the source term to the conservative flux term, then an equation that moves some (or all) of the gravitational potential from the source to the flux term should be equally valued. If this argument is accepted, then the preferred differential form for natural channels with the SVE is none of those presented above, but perhaps an equation of the general form

$$\frac{\partial Q}{dt} + \frac{\partial}{\partial x}\left\{\beta\frac{Q^2}{A} + f_1\left(A, \eta\right)\right\} = f_2\left(A, \eta\right) - gAS_f \tag{9}$$

where $f_1$ and $f_2$ are some functions (as yet unspecified) of the free-surface elevation rather than the depth. The free surface ($\eta$) can be interpreted as the uniform piezometric pressure over cross-section $A$, so the $f_1$ and $f_2$ functions are a generic way of splitting the piezometric pressure gradient between the source term and the conservative flux term, similar to how Cunge-Liggett handles the hydrostatic pressure gradient. Note that this proposed general approach removes the need for specifying $S_0$, which is important both in terms of source-term smoothness and producing "well-balanced" methods as discussed in detail in Section 3, below.

In this paper, complete derivations are presented to show that finite-volume formulations of the SVE can be generated that are consistent with the general conservative differential form of eq. (9). The new derivation is intended to bridge the gap between approaches used in high-resolution hydraulic models and those used in large-scale hydrology and urban drainage networks. The derivation provides a form of the SVE that has mathematical rigor while preserving the simplicity of the non-conservative finite-difference discretizations that are common in hydrological and urban drainage literature. Herein, we focus only on the detailed presentation of the new equation form, reserving demonstration in a numerical model to future papers.

## 3 Background

### Origination and use of the SVE

Alexandre de Saint-Venant's equations (de Saint-Venant, 1871) were written as:

$$\frac{dA}{dt} + \frac{d\left(AU\right)}{dx} = 0 \tag{10}$$

$$-\frac{d\eta}{dx} = \frac{1}{g}\frac{dU}{dt} + \frac{U}{g}\frac{dU}{dx} + \frac{\ell_p}{A}\frac{F}{\rho g} \tag{11}$$

where $U$ is the velocity, $\ell_p$ is the wetted perimeter, and $F$ is the frictional force per unit bottom area along the channel, and other terms are as previously noted. We have taken the liberty of replacing Saint-Venant's notation of $w, \zeta, \chi, s$ with the more

modern nomenclature of $A, \eta, \ell_p, x$, but otherwise have retained the original form. The momentum equation of de Saint-Venant, eq. (11), is identical to eq. (1) if we use $Q = AU$, $S_f = \ell_p F/(g\rho A)$, integrate over a cross-section with the $\beta$ coefficient, apply some calculus with the continuity equation, and define $\eta \equiv H + z_b$ along with $S_0 = -\partial z_b/\partial x$. From a practical perspective, the only thing that a hydrologist really needs to change from the original equation set is to replace the zero on the right-hand-side of
eq. (10) with a source term representing the inflow/outflow per unit length from/into the catchment and groundwater. However, from a numerical modeling perspective, eq. (11) is fundamentally non-conservative and suitable only for discretization in finite-difference forms. Although the full equation set is sometimes called the SVE, for convenience in exposition we will use SVE as a shorthand for the momentum equation alone.

   The SVE are ubiquitous in the literature for a wide range of work and have a foundational role in flow routing schemes
in hydrological models and channel network models for urban drainage. However, there is an interesting gap between the equation forms used in large-scale systems and those used in shorter single-reach studies or modeling hydraulic features. For computational simplicity, large-scale network flow models often use a reduced set of equations, such as Muskingum, kinematic wave, or the local inertia form (e.g. Wang et al., 2006; David et al., 2011, 2013; Getirana et al., 2017). Herein, we will follow the arguments of Hodges (2013) that we should be using the full SVE; i.e., reduced-physics models should be seen as a stop-
gap measure as we wrestle with obtaining satisfactory SVE solution methods. As computational power has increased, our large-scale models have been moving towards the full SVE but typically in a non-conservative form (e.g. Paiva et al., 2013; Liu and Hodges, 2014). For urban drainage networks, the US EPA Storm Water Management Model (SWMM) and variants built on this public domain model use a non-conservative finite-difference form of the SVE. This model is widely applied (e.g. Gulbaz and Kazezyilmaz-Alhan, 2013; Hsu et al., 2000; Krebs et al., 2013), however deficiencies in conservation are a
recognized problem (Rossman, 2017) and the SVE solver is the critical computational expense in the modeling system (Burger et al., 2014). Engineering river hydraulics problems are often solved using the U.S. Army Corps of Engineers HEC-RAS software, which has free model executables with a proprietary (closed-source) code base. HEC-RAS uses a non-conservative finite-difference form of the SVE based on methods pioneered in last quarter of the 20th century (Brunner, 2010). In contrast, more recent research models of the SVE at short river-reach scales have typically used the equations presented as hyperbolic
conservation laws that ensure both conservation and well-behaved solutions for subcritical, supercritical, and transcritical flows (e.g. Guinot, 2009; Ivanova et al., 2017; Papanicolaou et al., 2004; Sanders et al., 2003).

   There is also a vast gulf between the spatial discretization of SVE for large systems and smaller system studies in the hydraulics and applied mathematics literature (although the gap is getting narrower). For example in 2003 the SVE were solved at 1 to 4 km spacing for 156 km of river (Saavedra et al., 2003). Seven years later we find 4 km cross-section spacing
for $5 \times 10^3$ km of river (Pramanik et al., 2010). By 2014 the state-of-the-art was 100 m spacing for $15 \times 10^3$ km of river (Liu and Hodges, 2014). In contrast, hydraulic studies have typically focused on 1 to 10 m spacing for 1 to 2 km test cases (e.g. Gottardi and Venutelli, 2003; Kesserwani et al., 2009; Sart et al., 2010; Venutelli, 2006). Between these extremes, single-reach river models with natural geometry are typically modeled over river lengths less than 20 km with grid cells on the order of 10 m to more than 100 m (Sanders et al., 2003; Catella et al., 2008; Castellarin et al., 2009; Lai and Khan, 2012).

**Preissmann *v.* Godunov**

Computational modeling of the SVE is arguably a long-running contest between the differential, finite-difference governing equations pioneered Preissmann (1961) and the integral, finite-volume formulations derived from Godunov (1959). Clearly this is a simplification as there are wide-ranging contributions across both numerical methods and implementation schemes –

5   but the literature is simply too broad to discuss all developments in anything less than a book. Nevertheless, using a dialectic of Preissmann *v.* Godunov is a useful way of thinking about the major developments and provide context for the equations derived herein. Preissmann developed effective finite-difference methods applied to the non-conservative form of the equations and is the first basis for comparison of succeeding finite-difference models. The introduction of Roe's approximate Riemann solver (Roe, 1981) and analyses of Harten et al. (1983) made Godunov-like methods tractable and set off a multi-decadal

10  development of finite-volume methods. The literature with these two methods is vast, but a reasonable cross-section is provided in Table 1. Beyond these two major families, a variety of other schemes have been applied including finite-element methods (e.g., Szymkiewicz, 1991; Venutelli, 2003), finite-volume methods that do not use the Godunov approach (e.g. Audusse et al., 2004, 2016; Catella et al., 2008; Katsaounis et al., 2004; Mohamed, 2014; Vazquez-Cendon, 1999; Xing and Shu, 2011; Ying et al., 2004), and finite-difference methods that do not apply the Preissmann scheme (e.g., Abbott and Ionescu, 1967; Arico and Tucciarelli, 2007; Buntina and Ostapenko, 2008; Schippa and Pavan, 2008; Tucciarelli, 2003; Wang et al., 2000). A recent

development is the introduction of Discontinuous Galerkin (DG) methods, which can be thought of as a higher-order Godunov method (e.g., Kesserwani et al., 2008, 2009; Lai and Khan, 2012; Xing, 2014; Xing and Zhang, 2013).

It is clear from the above that there is no consensus on the best method for solving the SVE. For high-resolution modeling that correctly preserves shocks and transcritical flows, it can be reasonably argued that finite-volume and DG schemes are more successful than finite-difference schemes. Beyond that broad observation, the question of whether a finite-volume method

with Godunov-like approach is better than non-Godunov approach does not have a clear answer either in terms of accuracy or computational run times. However, in terms of large-scale systems the Preissmann scheme and finite-difference methods presently reign supreme with the ability to solve more than $15 \times 10^3$ km of river on a desktop computer (Liu and Hodges, 2014). Nevertheless, we can see that a conservative finite-volume approach for large-scale systems would be preferred as the basis for simulating Continental River Dynamics (Hodges, 2013) as well as for the challenges of urban drainage modeling

for the mega-cities that are growing across the earth. Being able to control numerical dissipation of momentum and ensure conservative fluxes will be increasingly important as advancing computing power pushes down the practical model grid scales.

**The "well-balanced" problem and $S_0$**

That finite-volume solutions are not commonly used in large-scale hydrologic and urban drainage models is a testament to their complexity. The difficulties associated with finite-volume solutions using the Cunge-Liggett conservative form of the

SVE have engendered a broad literature on "well-balanced" schemes deriving from the study of Greenberg and Leroux (1996). A principle feature of a well-balanced scheme is that it provides exactly steady solutions for exactly steady flows. The most trivial requirement (which is often not met) is that a flat free surface should result in exactly zero velocities. This problem is

**Table 1.** Two decades of Preissman v. Godunov

| Preissman | Godunov |
|---|---|
| Canelon (2009) | Bollermann et al. (2013) |
| Casas et al. (2010) | Delis and Skeels (1998) |
| Castellarin et al. (2009) | Delis et al. (2000b) |
| Chau and Lee (1991) | Delis et al. (2000a) |
| Chen et al. (2005) | Ferreira et al. (2012) |
| Gasiorowski (2013) | Gottardi and Venutelli (2003) |
| Islam et al. (2008) | Goutal and Maurel (2002) |
| Leandro and Martins (2016) | Greenberg and Leroux (1996) |
| Liu and Hodges (2014) | Guinot (2009) |
| Lyn and Altinakar (2002) | Ivanova et al. (2017) |
| Paiva et al. (2011, 2013) | Kesserwani et al. (2010) |
| Paz et al. (2010) | Kurganov and Petrova (2007) |
| Rosatti et al. (2011) | Li and Chen (2006) |
| Saavedra et al. (2003) | Liang and Marche (2009) |
| Sart et al. (2010) | Monthe et al. (1999) |
| Saleh et al. (2013) | Pu et al. (2012) |
| Sen and Garg (2002) | Sanders (2001) |
| Venutelli (2002) | Sanders et al. (2003) |
| Wu et al. (2004) | Venutelli (2006) |
| Zeng and Beck (2003) | Wu and Wang (2007) |
| Zhu et al. (2011) | Ying and Wang (2008) |

readily illustrated by considering eq. (4) with $Q = 0$, which implies $S_f = 0$. Achieving the simple result of a flat free surface for $Q = 0$ with the Cunge-Liggett form requires

$$\frac{\partial I_1}{\partial x} = I_2 + A S_0 \iff H + z_b = \text{constant} \tag{12}$$

which implies the Cunge-Ligget form is only well-balanced if the geometry meets the following identity at every possible water surface level:

$$\frac{\partial}{\partial x} \int_{z_b(x)}^{z_R} (z_R - z) B \, dz - \int_{z_b(x)}^{z_R} (z_R - z) \frac{\partial B}{\partial x} \, dz = A S_0 \quad : \quad z_b < z_R \leq \eta_{max} \tag{13}$$

where $\eta_{max}$ is the maximum water surface elevation, $z_b(x)$ is the local channel bottom elevation, and $z_R$ encompasses all pos-

sible water surface elevations. Clearly, designing a numerical scheme that *exactly* preserves this relationship for non-uniform

channels is a challenge, as evidenced by the breadth and complexities of studies focused on this issue (e.g., Audusse et al., 2004; Bollermann et al., 2013; Bouchut and Morales de Luna, 2010; Castro Diaz et al., 2007; Crnković et al., 2009; Kesserwani et al., 2010; Kurganov and Petrova, 2007; Li et al., 2017; Liang and Marche, 2009; Perthame and Simeoni, 2001; Xing, 2014). Failure to satisfy the well-balanced criteria results in models that generate spurious velocities; i.e., a mismatch in eq. (13) indicates that the numerics provide momentum sources/sinks that are functions of channel shape and discretization rather than flow physics. An interesting approach to this problem was developed by Schippa and Pavan (2008) where eq. (12) is used to replace $I_2 + AS_0$ in the source term with $\partial I_1/\partial x$ evaluated for a horizontal surface. Their approach ensures that *any* discretization will be well-balanced for a zero-velocity flow.

The work of Schippa and Pavan (2008) and the review of other works on well-balanced schemes provides us a key insight: the principal challenge for obtaining a well-balanced method is the channel bottom slope, $S_0$, which is often sharply varying or even discontinuous in a natural system. Furthermore, as a geometrical property, $S_0$ should be independent of the cross-sectional flow area ($A$), and yet is forced to be discretely related through eq. (12). If we take this idea a step further, we can argue that the fundamental problem with the Cunge-Liggett form is that the physical forces that alter momentum (gravitational potential and hydrostatic pressure) are arbitrarily separated so that one is wholly within the source term and the other has an *ad hoc* split between conservative flux and source terms. Thus, we return to the idea put forward in the Introduction that we should consider the free-surface elevation (piezometric pressure) instead of the water depth (hydrostatic pressure) as our primary forcing gradient.

In the next section, we shall see how the idea of shifting portions of the total piezometric pressure from source to flux can be used to develop a rigorous, conservative, and well-balanced finite-volume form of the SVE that is simpler than those based on the Cunge-Liggett form.

## 4 Finite-volume SVE with minimal approximations

**Continuity**

Although we are focused on the momentum equation, for completeness we will start with continuity. The general arrangement of the control volume for an irregular channel and the vectors used in the following discussion are illustrated in Fig. 1. Applying only the incompressibility approximation for a uniform density fluid, the volume-integrated continuity equation is

$$\frac{\partial V}{\partial t} + \oint_S u_k n_k \, dA = S_V \tag{14}$$

where the Einstein summation convention is applied on repeated subscripts, $u_k$ is a vector velocity, and $n_k$ is a unit normal vector defined as positive pointing outward from a control volume $V$, and $S_V$ is a volume source ($S_V < 0$ for a sink).

A semi-discrete finite-volume representation of continuity can be directly written as

$$\frac{\partial V_e}{\partial t} = Q_u - Q_d + q_e L_e \tag{15}$$

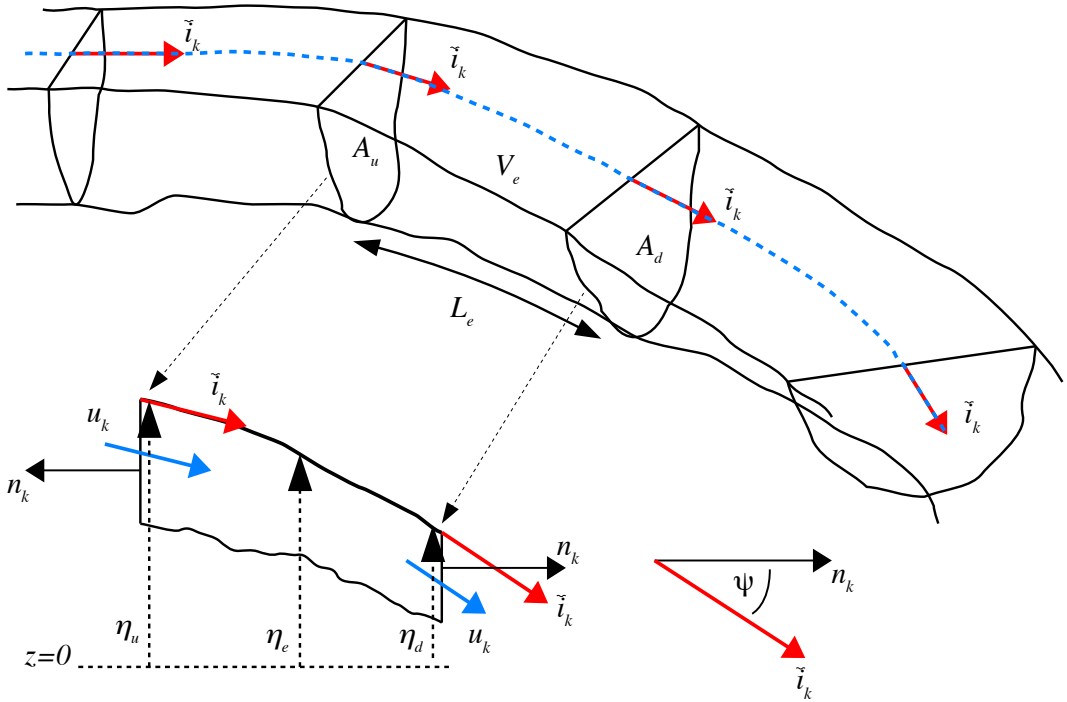

**Figure 1.** General arrangement of control volume element ($V_e$) and its neighbors for irregular channel. Unit normal vectors $n_k$ are always perpendicular to cross-sectional areas ($A_u$, $A_d$) and pointing outward from control volume. The element length ($L_e$) is measured along the channel. Unit vectors $\hat{i}_k$ are coincident with the free surface slope in the streamwise direction and can be defined as local continuous functions. The velocity vector $u_k$ is approximated as parallel to $\hat{i}_k$. The angle measured from $n_k$ to $\hat{i}_k$ is $\psi$. The free surface elevations $\eta_u$, $\eta_e$, and $\eta_d$, are cross-section uniform elevations at the upstream face, for the element center, and the downstream face, respectively. Note that volume is $V$ throughout the derivations with $u$ and $U$ used for continuous and spatially-averaged velocities, respectively.

where $Q$, and $L$ represent the flow rate and element length, and subscripts $e, u,$ and $d$ denote characteristic values for the control volume element, nominal upstream face, and nominal downstream face, respectively. Here we use the "nominal" flow direction as the global downhill direction of the channel that is assigned at the network level. The average lateral inflow per unit length is $q_e$, and a flow $Q > 0$ is from the upstream to downstream direction. Reversals of flow from the nominal flow direction are handled with $Q < 0$.

## Momentum

The control-volume form of the Navier-Stokes momentum equation in a direction defined by unit vector $\hat{i}$ in a Cartesian frame is

$$\frac{\partial}{\partial t} \int_V u_{\hat{i}} dV + \oint_S u_{\hat{i}} u_k n_k \, dA + \frac{1}{\rho} \oint_S p \hat{i}_k n_k \, dA = \oint_S \nu \frac{\partial u_j}{\partial x_k} \hat{i}_j n_k \, dA + \int_V g_{\hat{i}} dV \tag{16}$$

where $u_{\hat{i}}$ is a velocity vector component in the $\hat{i}$ direction (i.e., the direction that is *a priori* of interest), $u_k$ are velocity components along Cartesian axes, the component of the gravity vector in the $\hat{i}$ direction is $g_{\hat{i}}$, the kinematic viscosity is $\nu$, and $p$ represents the thermodynamic pressure. Note that this formulation can be related to any arbitrary Cartesian axes. In many common derivations, $\hat{i}$ is approximated as coincident with an $x$ axis that is a *horizontal* vector in the streamwise direction. In the following, we will show that this approximation is not required. Instead, we treat this as a simplification that can be applied to the final equation form.

## Advection terms

The $\hat{i}$ direction for momentum, eq. (16), is a vector associated with the $u_{\hat{i}}$ velocity component, which is not necessarily coincident with the normal vector $n_k$ at a flux surface of a finite volume (in contrast to the case where $\hat{i}$ is taken as horizontal). For a gradually-varying open-channel flow we can take the $\hat{i}$ vector as the nominal downstream direction along the channel centerline described by a vector *that lies along the free surface*, as illustrated in Fig. 1. Thus, this vector is local (as opposed to being forced into coincidence with a Cartesian axis) and changes along the channel with the slope of the free surface. It follows that a discrete control-volume formulation developed from eq. (16) can be globally exact as $V_e \to 0$. In contrast, a derivation that takes $\hat{i}$ as a vector in the horizontal direction has a momentum conservation error proportional to $\cos\psi$, where $\psi$ is an angle between the horizontal vector and the free surface; such an error does not vanish as $V_e \to 0$ unless the free surface is flat across the length of the element. This idea helps illustrate one of the subtle implications of the Godunov approach in which the channel is imagined as having a piecewise flat free surface: $\cos\psi = 1$ is then an identity within the approximation of the physics rather than as an approximation within the mathematics.

For equation (16), the upstream element face is required to be vertical and normal to the smooth channel centerline in a horizontal plane, as illustrated in Fig. 1. The free surface at the centerline has an angle of $\psi(x)$ to the horizontal so that the discrete nonlinear momentum term in the $\hat{i}$ direction across the upstream face is formally

$$\int_{A_u} u_{\hat{i}} u_k n_k dA = (\beta Q U_{\hat{i}})_u \tag{17}$$

where $Q$ is the flowrate, $U_{\hat{i}}$ is the average streamwise velocity over the cross-sectional area, the $u$ subscript indicates the upstream face (rather than vector components), and $\beta$ is the momentum coefficient for the streamwise velocity, defined as

$$\beta \equiv \frac{1}{A\left(U_{\hat{i}}\right)^2} \int_A \left(u_{\hat{i}}\right)^2 dA \tag{18}$$

Note that the only approximation in the convective term of eq. (17) is that the streamwise velocity is parallel to the free surface. However, the interpretation of the $u_{\hat{i}}$ and $u_k n_k$ terms may not be obvious, so further explanation is provided in Appendix A. For notational convenience, it is useful to let $U \equiv U_{\hat{i}}$. However, since $Q = A \int u_k n_k \, dA$, strictly speaking this requires an unconventional $Q = AU \cos\psi$. If the channel is straight and the free surface is linear so that the upstream face is parallel to the downstream face and there is a single value of $\psi$, it follows that an exact finite-volume integration of the nonlinear advection term is

$$\oint_S u_{\hat{i}} u_k n_k \, dA = -\beta_u Q_u U_u + \beta_d Q_d U_d - M_e \tag{19}$$

where $M_e$ represents any integrated sources (+) and sinks (−) of momentum per unit mass in the finite-volume element associated with the $q_e L_e$ lateral fluxes of eq. (15). Note that $M_e$ terms are typically neglected in SVE solvers. For the more general case where the channel is curved between the upstream and downstream faces and the free surface gradient changes (as in Fig. 1), the above integration becomes an approximation that is only exactly satisfied in the limit as the control volume length goes to zero. For present purposes, the use of a gradually-varying streamwise $\hat{i}$ direction implies that pressure is perfectly redirecting momentum through bends and aligning the momentum with the free surface. These are (generally) unstated approximations used in common 1D SVE formulations. However, it should be noted that this perfect momentum redirection is not precisely correct; e.g., secondary circulation in bends affects bed shear, velocity distribution, and frictional losses (Blanckaert and Graf, 2004). Arguably, losses associated with flow redirection in channel bends and/or rapid changes in the free surface gradient should be built into the frictional term in any model. Unfortunately, this remains a relatively poorly-studied area at the interface of hydrology and hydraulics. The curvature effects on the equations can be written as perturbation terms that relate the channel width to the radius of curvature (Hodges and Imberger, 2001; Hodges and Liu, 2014), but these ideas have yet to be exploited in developing curvature effects in SVE numerical models.

**Pressure decomposition**

For the pressure term in eq. (16) we follow the traditional approach for incompressible flows of defining a modified pressure ($\tilde{P}$) that includes the gravity term, which requires $\tilde{P} = p + \rho g z$. More formally we define

$$\frac{1}{\rho} \oint_S \tilde{P} \hat{i}_k n_k \, dA \equiv \frac{1}{\rho} \oint_S p \hat{i}_k n_k \, dA - \int_V g_{\hat{i}} \, dV \tag{20}$$

The surface integral for the modified pressure decomposes into integrations over the upstream cross-section ($A_u$), the downstream cross-section ($A_d$), the channel bottom ($A_B$), and the free surface ($A_\eta$) as:

$$\frac{1}{\rho} \oint_S \tilde{P} \hat{i}_k n_k \, dA = \frac{1}{\rho} \left\{ \int_{A_u} \tilde{P} \hat{i}_k n_k \, dA + \int_{A_d} \tilde{P} \hat{i}_k n_k \, dA + \int_{A_B} \tilde{P} \hat{i}_k n_k \, dA + \int_{A_\eta} \tilde{P} \hat{i}_k n_k \, dA \right\} \tag{21}$$

Note that $A_B$ includes the both the bottom and side walls (if any). The above is an exact pressure term without imagining the geometry to be rectilinear or that the free surface has any simplified shape. The only geometric requirement of the volume is

that faces $A_u$ and $A_d$ must be vertical planes that cut across the channel, which is necessary for consistency with the convective term. The first simplification we can introduce is to approximate the free surface as uniform at any cross section. The slope of the free surface is then everywhere coincident with the $\hat{i}(x)$ vector, which is aligned with the free surface at the channel centerline. With the free surface aligned with $\hat{i}(x)$, the modified pressure is everywhere normal to the free surface and cannot contribute to the streamwise momentum (which we have defined as parallel to $\hat{i}(x)$ in deriving the advection terms, above). It follows that $\tilde{P}\hat{i}_k n_k$ is identically zero at the free surface and the last term in eq. (21) vanishes.

**Piezometric and non-hydrostatic pressure**

It will be convenient to introduce a decomposition of the modified pressure ($\tilde{P}$) into a piezometric pressure ($P$) and non-hydrostatic pressure ($\breve{P}$), where only the latter is non-uniform over a cross-section. The non-hydrostatic pressure is defined by the difference between the modified pressure and the piezometric pressure:

$$\breve{P}(x,y,z) \equiv \tilde{P}(x,y,z) - P(x) \tag{22}$$

The piezometric pressure is formally the sum of the hydrostatic pressure and the gravitational potential at any point $z$ for $z_b \le z \le \eta$, which provides $P(x,y,z) \equiv \rho g\left[\eta(x,y) - z\right] + \rho g z = \rho g\eta(x,y)$. For the SVE, the free-surface elevation can be considered uniform over the channel breadth (i.e., neglecting cross-channel tilt in channel bends). It follows that the piezometric pressure is

$$P(x) = \rho g\eta(x) \tag{23}$$

which is uniform over a vertical cross section. The non-hydrostatic pressure was neglected by Alexandre de Saint-Venant and arguably should be neglected in any 1D momentum equation bearing his name. However, for completeness we will retain the non-hydrostatic pressure terms in the derivation below, but it will be neglected in discrete forms of the piezometric pressure terms in Section 5.

**Pressure on flow faces**

To further simplify the first two pressure terms in eq. (21), we define $\psi(x)$ as the angle between a horizontal line and the free surface, $\eta(x)$, measured clockwise positive from the horizontal line pointing downstream. Thus, the $\hat{i}$ vector is generally at an angle $\pm\psi(x)$ and pointing in the nominal downstream direction, as shown in Fig. 1. At the upstream cross-section the pressure force without approximations is

$$\int_{A_u} \tilde{P}\hat{i}_k n_k \, dA = -\cos\psi_u \int_{A_u} \tilde{P}(x,y,z) \, dA \tag{24}$$

where subscript $u$ indicates values at the upstream cross section. Similarly, the downstream cross-section provides

$$\int_{A_d} \tilde{P}\hat{i}_k n_k \, dA = +\cos\psi_d \int_{A_d} \tilde{P}(x,y,z) \, dA \tag{25}$$

where subscript $d$ indicates values at the downstream cross section. Note that in the above and in the following derivations, the ubiquitous $1/\rho$ coefficient of all pressure terms will be omitted for clarity. Introducing the piezometric and non-hydrostatic pressure split of eq. (22) provides:

$$\int_{A_u} \tilde{P} \hat{i}_k n_k \, dA = -P_u A_u \cos \psi_u - \cos \psi_u \int_{A_u} \check{P}(x,y,z) \, dA \tag{26}$$

$$\int_{A_d} \tilde{P} \hat{i}_k n_k \, dA = +P_d A_d \cos \psi_d + \cos \psi_d \int_{A_d} \check{P}(x,y,z) \, dA \tag{27}$$

Note that using the piezometric pressure instead of the hydrostatic pressure ensures that the only term requiring discrete integration at the upstream and downstream surfaces is the non-hydrostatic pressure. That is, $PA\cos\psi$ is *not* an approximate
integral whose adequacy depends on simplifying assumptions in geometry (as is the case for integrals of hydrostatic pressure in the Cunge-Liggett form), but is instead an exact integration of piezometric pressure for *any* cross-section shape.

**Pressure on bottom topography**

The third pressure term in eq. (21) is more challenging than the pressure on the flow faces as the bottom surface normal ($n_k$) varies with irregular topography. Hence, the local piezometric pressure contribution in the streamwise direction ($\hat{i}$) at any
position $(x,y)$ depends not only on the local water surface elevation but on the local irregularities in topography. It follows that integrating this term over a control volume requires some approximation of the subgrid topography. To arrive at a simpler formulation we note that the pressure acting normal to an arbitrary topography element at position $(x,y)$ with surface normal $n_k$ will have force components that can be resolved along a set of local Cartesian axes. One axis is taken along a topography slope angle of $\theta$ that lies in the same vertical plane as the streamwise direction $\hat{i}$, as illustrated in Fig. 2. We can imagine
Fig. 2 as an infinitesimal slice across a channel of irregular topography such that a series of these slices (with different $\theta$) can represent any cross-section shape. The second local Cartesian axis is taken within the same vertical plane but perpendicular to the axis defined by $\theta$. The third axis, perpendicular to the other two, is necessarily horizontal and in the cross-channel direction (out of the page in Fig. 2). We are interested only in how the topography contributes to pressure in the streamwise $\hat{i}$ direction, so by definition the cross-channel pressure component is irrelevant. This approach is entirely consistent with changes in depth across the channel and changes in breadth along the channel – both merely alter the surface normal vector $n_k$ that is resolved into the local Cartesian system based on the local slope direction of $\theta$.
5 To isolate the pressure forces acting in the $\hat{i}$ streamwise direction, as a conceptual model we can imagine the topography in the infinitesimal slice of Fig. 2 replaced with a set of $m = 1...N$ stair-steps, where the treads are locally parallel to the free surface and the risers are normal to the free surface, as illustrated in Fig. 3. Clearly, as $N \to \infty$ we will recover a continuous approximation so there is no need to actually consider the discrete stair-steps in a solution method – the steps are merely to illustrate what is otherwise a mathematical abstraction in vector calculus. As illustrated in Fig. 4, we can imagine the stair-step risers as thin planar strips across the entire wetted perimeter that provide a discrete representation of irregular channel cross-section structure. The only requirements for this conceptual model are that the free surface at longitudinal position $x$

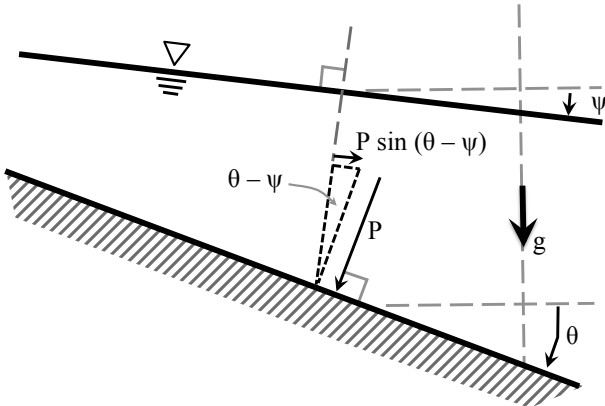

**Figure 2.** Pressure decomposition to obtain streamwise contribution.

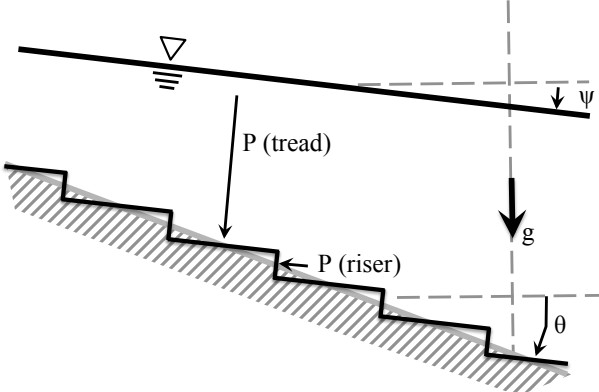

**Figure 3.** Stair-step approximation for pressure along bottom where tread is parallel to the free surface and riser is perpendicular for linear slopes of both bottom and free surface.

is uniform over the cross-section and the slope is aligned with $\hat{i}(x)$. This conceptual model could also be envisioned in 3D

5    as discrete rectilinear bricks that approximate the topography: the upper surface of the brick is always aligned parallel with the free surface, the side face is across the channel, and only the front (or back) face is perpendicular to $\hat{i}$ and contributes a topographic pressure force in the streamwise direction. As the brick dimensions go to zero the continuous topography is obtained.

     Since the stair-step treads are (by definition), normal to the modified pressure above, it follows that the only pressure contri-

10    butions to the momentum in the streamwise $\hat{i}$ direction are on the risers, with individual areas $A_{R(m)}$ for $m = 1...N$ stairsteps. Because the pressure contribution for increasing cross-sectional area (i.e., steps down as in Fig. 3) will be opposite of the pressure contribution for decreasing cross-sectional area (i.e., steps upward), it is convenient to introduce a function $\gamma_{(m)} = \pm 1$ to

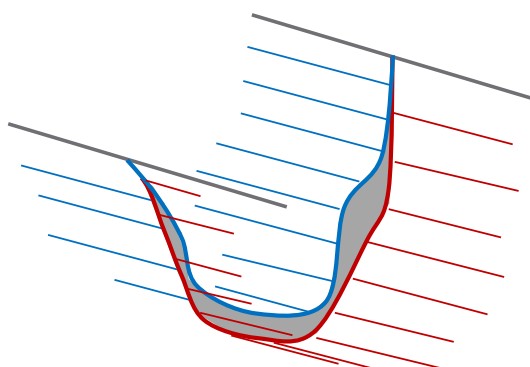

**Figure 4.** Three-dimensional conceptual model of stair-step riser (gray) as a 2D planar area of $A_{R(m)}$ separating two tread areas (blue, red).

account for the change of sign needed for the direction of the pressure force. We can formally define $\gamma_{(m)} \equiv \hat{i}_k \hat{j}_k$ at step $m$ where $\hat{j}_k$ is the normal unit vector (pointing outwards) from the $A_{R(m)}$ riser, as shown in Figs. 5 and 6 for two different nonlinear water surface profiles over identical bottom topography. It follows that

$$\int_{A_B} \tilde{P}\hat{i}_k n_k \, dA \approx -\sum_{m=1}^{N} \gamma_{(m)} \int_{A_{R(m)}} \tilde{P}(x,y,z) \, dA \tag{28}$$

The above summation can be written as an integral over the length $L$ of the finite-volume element as $N \to \infty$.

$$\int_{A_B} \tilde{P}\hat{i}_k n_k \, dA = -\int_L \gamma(x) \int_{A_R(x)} \tilde{P}(x,y,z) \, dA \, dx \tag{29}$$

where $\gamma(x) = \hat{i}_k \hat{j}_k$ is the continuous counterpart to the discrete $\gamma_{(m)}$. Note that this conceptual model is valid even for non-monotonic behavior of the riser area (e.g., Fig. 6) as long as the $A_{R(m)}$ are continuous and smooth as $N \to \infty$. However as discussed in Section 5, below, extremes of non-monotonic behavior can make it difficult to create a consistent discrete equation for the topographic pressure for a control volume of finite size.

For further simplification, it is convenient to introduce the piezometric/non-hydrostatic splitting of eq. (22) where, by definition, the piezometric pressure is is uniform over a vertical cross section (e.g., a stair-step riser). Let $P_{(m)}$ represent the piezometric pressure at the $m$ stair-step riser with area $A_{R(m)}$ so that over a control volume for steps $m = 1...N$. It follows that

$$\sum_{m=1}^{N} \int_{A_{R(m)}} P_{(m)} \, dA = \sum_{m=1}^{N} P_{(m)} \int_{A_{R(m)}} dA$$

$$= \int_L P(x) A_R(x) \, dx \quad \text{for} \quad N \to \infty \tag{30}$$

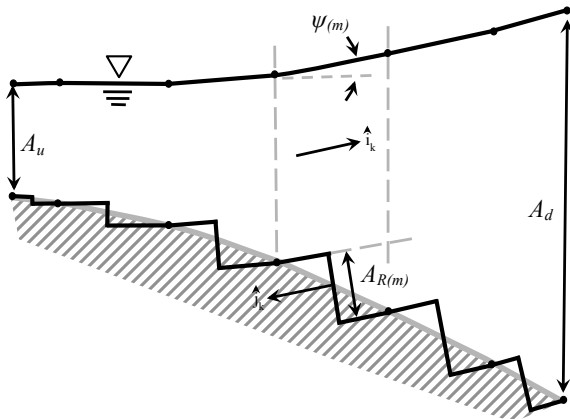

**Figure 5.** Conceptual model of piecewise linear approximation of nonlinear free surface and stair-step approximation of nonlinear topography with the cross-sectional area monotonically increasing. Pressure on the discrete riser area $A_{R(m)}$ is locally aligned with free surface slope directly above. The vertical scale and thus the tilt of $A_{R(m)}$ are exaggerated for illustrative purposes.

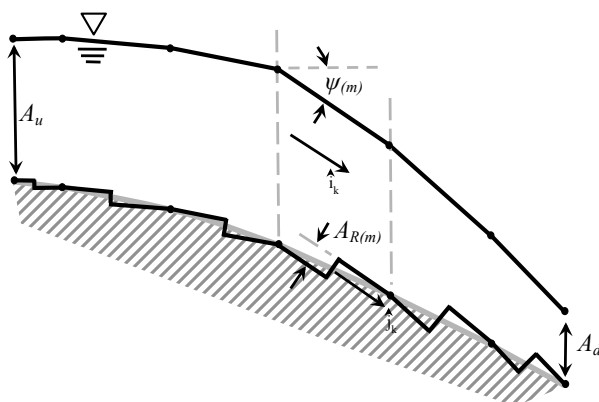

**Figure 6.** Conceptual model of piecewise linear approximation of nonlinear free surface and stair-step approximation of nonlinear bottom with cross-sectional area increasing from $A_u$ to the center and then decreasing from the center to $A_d$. Pressure on the discrete riser area $A_{R(m)}$ is locally aligned with free surface slope directly above. The vertical scale and thus the tilt of $A_{R(m)}$ are exaggerated for illustrative purposes. Note that the topography is identical to Fig. 5 but the stair steps are different due to the alteration of the free surface.

where the continuous $A_R(x)$ represents the effective bottom area contributing to the piezometric pressure force in the stream-

wise $\hat{i}(x)$ direction. Applying this continuous $A_R(x)$ form and the hydrostatic/nonhydrostatic splitting to eq. (29), we obtain

$$\int\limits_{A_B} \tilde{P}\hat{i}_k n_k\, dA = -\int\limits_L \gamma(x)P(x)A_R(x)\, dx - \int\limits_L \gamma(x)\int\limits_{A_R} \breve{P}(x,y,z)\, dA\, dx \tag{31}$$

which is a complete representation of the topographic pressure contribution to streamwise momentum in terms of the effective riser areas – i.e., the contribution based on the component of the bottom normal projected in the streamwise direction.

The stair-step conceptual model and $A_R$ allow us to consider the pressure effects along the $\hat{i}$ direction due to the changing cross-sectional area of the channel without introducing the separate force terms $I_1$ and $I_2$ of the Cunge-Liggett form of the SVE. An interesting part of this model is that $A_R(x)$ over a control volume is a function of *both* the local bottom topography and the local free surface slope. That is, comparing Fig. 5 and 6 we see that $A_{R(m)}$ riser areas are different, despite the identical bottom topography. Returning to our idea that 3D topography can be represented by discrete bricks, we can imagine each brick is pinned on an axis that allows it to locally rotate to different angles so that the upper surface is always parallel to the free surface. Again, the continuous topography is recovered as the brick size goes to zero, but the brick rotation allows the representation of the different topography effects that are caused by the interaction between the change in relationship between the downstream vector, $\hat{i}$, and the bottom normal vector as the free-surface profile evolves. Thus, $A_R$ is a dynamic representation of the interaction between the free-surface gradient and bottom topography that controls the effective along-stream pressure gradient of converging or diverging flow areas.

In theory, we might directly compute $\int P A_R \, dx$ over a control volume; however, it seems likely that direct discretization of subgrid topography could cause unbalanced momentum source terms. In effect, computing $\int P A_R \, dx$ has the same complexity as computing the $I_1$ or $I_2$ terms of the Cunge-Liggett form and gains us little. Thus, it is useful to consider limiting approximations that can be developed from examining the geometry of Figs. 3 and 5. In the simplest case where both the free surface and bottom have linear slopes (Fig. 3), geometry for the infinitesimal slice requires that

$$A_d \cos \psi - A_u \cos \psi = \sum_{m=1}^{N} A_{R(m)} \tag{32}$$

where $A_d \cos \psi$ represents the cross-sectional area normal to the free surface at the downstream cross section and there is only one value of $\psi$ over the control volume for the linear free-surface slope. Note that the above is an identity for any discrete $N$ stairsteps for the linear case. The geometry is somewhat more complex for the nonlinear system of Fig. 5, but the $m$ stair step with free surface angle $\psi_{(m)}$ must satisfy a similar geometric identity

$$A_{(m-1/2)} \cos \psi_{(m)} - A_{(m+1/2)} \cos \psi_{(m)} = A_{R(m)} \tag{33}$$

where $A_{(m\pm1/2)} \cos \psi_{(m)}$ are the areas normal to the free surface on the upstream and downstream edges of the $m$ piecewise linear stair step. For the nonlinear free surface and/or topography over adjacent linear stair-steps there is a discontinuity of the treads for adjacent steps as $\cos \psi_{(m-1)} \neq \cos \psi_{(m)}$, so the discrete summation of the stair step areas over a control volume provides an approximation rather than an identity:

$$A_d \cos \psi_d - A_u \cos \psi_u \approx \sum_{m=1}^{N} A_{R(m)} \tag{34}$$

However, in the limit as $N \to \infty$ we have a single value of $\cos \psi$ at any point along a smooth free surface so that the continuous form provides an identity:

$$A_d \cos \psi_d - A_u \cos \psi_u = \int_L A_R(x)\, dx \tag{35}$$

To generalize the above for $A_d < A_u$, we can use the $\gamma(x) = \pm 1$ that was introduced for the pressure direction in eq. (29). Values of $\gamma(x) = +1$ indicates the cross-section area is increasing across location $x$ in the streamwise direction (as in Figs. 3 and 5), whereas $\gamma(x) = -1$ indicates the cross-sectional area is decreasing (as in the latter portion of Fig. 6). It follows that

$$A_d \cos \psi_d - A_u \cos \psi_u = \int_L \gamma(x) A_R(x)\, dx \tag{36}$$

is an identity that should be satisfied for any control volume where the bottom topography and free surface are continuous and smooth. Note that if Fig. 3 is imagined as one of many infinitesimal slices (with varying $\theta$) that make up a channel cross section, it should be obvious that Eq. (36) also applies for a finite volume with irregular topography.

To handle the integration of $A_R(x)$ in the piezometric pressure term of eq. (31), we introduce a quadrature function $\lambda(x)$, defined as

$$\lambda(x) \equiv \frac{\gamma(x) A_R(x)}{A_d \cos \psi_d - A_u \cos \psi_u} \tag{37}$$

Note that with eq. (36), this implies the identity:

$$\int_L \lambda(x)\, dx = 1 \tag{38}$$

Using the above in the first term on the RHS of eq. (31), we obtain

$$\int_L \gamma(x)\, P(x)\, A_R(x)\, dx = (A_d \cos \psi_d - A_u \cos \psi_u) \int_L P(x)\, \lambda(x)\, dx \tag{39}$$

Thus, the introduction of $\lambda$ allows us to extract a multiplier from the control-volume integral of the bottom pressure. As a result, $\lambda(x)$ is merely a distribution, or "weighting" function for integration of $P(x)$. The full bottom pressure term, eq. (31), can be written as

$$\int_{A_B} \tilde{P} \hat{i}_k n_k\, dA = -(A_d \cos \psi_d - A_u \cos \psi_u) \int_L P(x)\lambda(x)\, dx - \int_L \gamma(x) \int_{A_R} \check{P}(x,y,z)\, dA\, dx \tag{40}$$

Note that $\lambda$ weighting cannot be readily applied the non-hydrostatic term because the non-hydrostatic pressure on the bottom has spatial distributions in both the vertical and across a channel that cannot be assumed negligible; hence we cannot pass $\check{P}$ through the $A_R$ integration as was done in eq. (30) for $P$.

We can think of $\lambda(x)$ as a weighting function of the conceptual stair-step riser areas over the control-volume length, which controls where the piezometric pressure gradients have their greatest effect. For example, in Fig. 3 the stair-step risers are

uniformly distributed such that we can use $\lambda(x) = L^{-1}$, which meets the identity requirement of eq. (38). In contrast, Fig. 5 implies $\lambda(x)$ is perhaps a quadratic function. Figure 6 presents a challenge as $\lambda(x)$ should reverse in sign between the upstream and downstream faces. A key point in this new finite-volume derivation is the selection of $\lambda$ functions provides a more

general discrete control over the representation of the free surface and bottom topography within a control volume. This can be contrasted to the Godunov approach that approximates a control volume as piecewise uniform for both the bottom elevation and free-surface elevation over a control volume (see Section 3). Several discrete approaches to approximation of $\lambda(x)$ will be examined in Section 5, although the full consequences and utility of the $\lambda$ approach will require more extensive investigation for both theoretical limitations and practical discretization schemes.

**Combining pressure terms**

In summary, the pressure terms of eq. (21) can be written using eqs. (26), (27) and (40), resulting in:

$$\frac{1}{\rho} \oint_S \tilde{P} \hat{i}_k n_k \, dA = -\frac{1}{\rho} A_u P_u \cos\psi_u + \frac{1}{\rho} A_d P_d \cos\psi_d - \frac{1}{\rho} \left( A_d \cos\psi_d - A_u \cos\psi_u \right) \int_L P(x)\lambda(x)\, dx$$

$$-\frac{\cos\psi_u}{\rho} \int_{A_u} \check{P}(x,y,z)\, dA + \frac{\cos\psi_d}{\rho} \int_{A_d} \check{P}(x,y,z)\, dA - \frac{1}{\rho} \int_L \gamma(x) \int_{A_R} \check{P}(x,y,z)\, dA\, dx \quad (41)$$

where the last three terms are the non-hydrostatic pressure effects that are typically neglected in the SVE.

**Viscous term**

The remaining term in eq. (16) is the viscous term, which is treated as an empirical function in all but the most highly-resolved models of simple systems – note that Decoene et al. (2009) provides a comprehensive and rigorous approach for friction that has not yet been fully considered in SVE models. For the present purposes, we will retain the simple friction slope form with an assumption of linear behavior over space, i.e.

$$\oint_S \nu \frac{\partial u_j}{\partial x_k} \hat{i}_j n_k \, dA = -g \int_{V_e} S_f(x)\, dV \approx -g V_e S_{f(e)} \quad (42)$$

where $S_{f(e)}$ is the average friction slope over the control volume $V_e$.

**Finite-volume for momentum**

Putting together the above, eq. (16) can be written in a finite-volume form as

$$5 \quad \frac{\partial}{\partial t}\left( U_e V_e \right) = \beta_u Q_u U_u - \beta_d Q_d U_d + \frac{1}{\rho} A_u P_u \cos\psi_u - \frac{1}{\rho} A_d P_d \cos\psi_d$$

$$+ \frac{1}{\rho} \left( A_d \cos\psi_d - A_u \cos\psi_u \right) \int_L P(x)\lambda(x)\, dx - \frac{\cos\psi_u}{\rho} \int_{A_u} \check{P}(x,y,z)\, dA + \frac{\cos\psi_d}{\rho} \int_{A_d} \check{P}(x,y,z)\, dA$$

$$- \frac{1}{\rho} \int_L \gamma(x) \int_{A_R} \check{P}(x,y,z)\, dA\, dx - g V_e S_{f(e)} + M_e \quad (43)$$

where $U_e$ is the element velocity in the streamwise direction, $V_e$ is the element volume and the relationship between $U$ and $Q$ is given by

$$Q = AU\cos\psi \tag{44}$$

Note that $Q > 0$ and $U > 0$ imply flow in the nominal downstream direction, whereas $Q < 0$ and $U < 0$ imply flow in the nominal upstream direction. At this point we have introduced only four approximations: (1) uniform-density incompressibility, (2) the effect of channel curvature is either negligible or handled in an empirical viscous term, (3) the cross-channel variability in the free-surface slope is negligible, and (4) a friction-slope model can be used to represented integrated viscous effects over a control volume. In addition we have a geometric restriction that the upstream and downstream control volume cross-sections must be vertical planes that are orthogonal (in the horizontal plane) to the mean flow direction.

For convenience in exposition, for the remainder of this paper we will apply the hydrostatic approximation ($\breve{P} = 0$) along with approximations for small slope ($\cos\psi \approx 1$), and uniform cross-section velocity ($\beta \approx 1$). Furthermore, we will limit our focus to flows without lateral momentum sources ($M_e = 0$). These simplifications allow us focus attention on the pressure source term, which is the primary new contribution in this derivation. The resulting simplification of eq. (43) can be presented with conservative terms on the LHS and source terms on the RHS as

$$\frac{\partial}{\partial t}(U_e V_e) - \frac{Q_u^2}{A_u} + \frac{Q_d^2}{A_d} - gA_u\eta_u + gA_d\eta_d = g(A_d - A_u)\int_L \eta(x)\lambda(x)\,dx - gV_e S_{f(e)} \tag{45}$$

where definition of piezometric pressure, eq. (23), is used to substitute $P = \rho g\eta$. A more formal finite-volume integral presentation would be

$$\frac{\partial}{\partial t}\int_V u\,dV - \int_{A_u} u^2\,dA + \int_{A_d} u^2\,dA - g\int_{A_u}\eta\,dA + g\int_{A_d}\eta\,dA = g(A_d - A_u)\int_L \eta(x)\lambda(x)\,dx - g\int_V S_f\,dV \tag{46}$$

Equation (45) can be reduced to a differential equation using $V = AL$ as $L \to 0$. Dividing through by $L$ and taking the limit as $L \to 0$ provides

$$\frac{\partial Q}{\partial t} + \frac{\partial}{\partial x}\left(\frac{Q^2}{A} + gA\eta\right) = g\eta\frac{\partial A}{\partial x} - gAS_f \tag{47}$$

If we substitute geometric identities $\eta \equiv H + z_b$ and $S_0 \equiv -\partial z_b/\partial x$, we see that the above becomes identical to eq. (8). Thus, our finite-volume derivation is exactly consistent with the commonly-used differential SVE that is posed using $S_0$. However, from eqs. (45) and (47) we see that the free-surface source term in the differential form, $g\eta\,\partial A/\partial x$, is related to the more interesting integral source term in the finite-volume form:

$$g(A_d - A_u)\int_L \eta(x)\lambda(x)\,dx$$

This term can be thought of as an integrated free-surface/topography term that reflects nonlinearity in both the free surface and topography over a control volume. It is clear that this finite-volume term collapses to $g\eta\,\partial A/\partial x$ as $L \to 0$ but the integral form is not readily inferred from the differential form. Through approximations of this integral term we can obtain a variety of different finite-volume forms of the SVE, as outlined in the following section.

## 5 Approximate finite-volume forms of the SVE

### General form

We are interested in approximate forms of the finite-volume SVE that arise from discretization choices in the integral source term derived above, so for exposition it will be convenient to start from the approximate form in eq. (45) with $V_e = A_e L_e$ and $Q_e = A_e U_e$. These approximations can be readily reversed to provide a more complete equation, but the simple form for further analysis is

$$L_e \frac{\partial Q_e}{\partial t} - \frac{Q_u^2}{A_u} + \frac{Q_d^2}{A_u} - gA_u\eta_u + gA_d\eta_d = T_e - gV_e S_{f(e)} \tag{48}$$

where $T_e$ is the source term for integrated free-surface/topography effects obtained from the RHS of eq. (45):

$$T_e \equiv g\left(A_d - A_u\right) \int_L \eta(x)\lambda(x)\,dx \tag{49}$$

Note that the LHS of eq. (48) is discretely conservative in that a summation over all elements will cause all the LHS terms to identically vanish except for the $\partial/\partial t$ and boundary conditions. Thus, the RHS are source terms, i.e. the traditional friction term and a term representing nonlinearities in the free surface interacting with topography can create and destroy momentum. The system is inherently "well-balanced" as discussed in Section 3, as long as $T_e$ identically vanishes for a flat free surface.

Equation (49) admits a wide variety of approximate equations, depending on the form chosen for the quadrature of $\eta(x)$ and $\lambda(x)$ over the length of a finite volume. Arguably, a simple finite-volume discrete method will have three values of $\eta$ that characterize a control volume: $\eta_u$, $\eta_e$, and $\eta_d$, as illustrated in Fig. 1. Different numerical methods can be constructed by using different approximations constructed from these three values. Herein, we cannot exhaustively investigate the variety of options and so will focus on the most obvious candidates, which are polynomials of orders 0 through 2. The zero-order polynomial is an approximation of $\eta(x)$ as a uniform value over the element length (e.g., as in the Godunov conceptual model, discussed in Section 3) – which could be simply represented by $\eta(x) = \eta_e$ so that the face values of $\eta_d$ and $\eta_u$ are ignored. Note that even this choice has alternate forms – a slightly different scheme could be constructed using $\eta(x) = (\eta_u + \eta_d)/2$, which is also uniform but ignores $\eta_e$. A first-order polynomial for $\eta(x)$ implies a linear free-surface slope across the element, which might be represented by a slope from $\eta_d$ to $\eta_u$. A second-order polynomial implies a quadratic curvature to the free surface that can pass smoothly through $\eta_u$, $\eta_e$, and $\eta_d$. Clearly this idea could be extended to cubic splines by including adjacent control volume values. Beyond these polynomials there are other options that might be suitable. For example, we could use the three discrete $\eta$ values to provide piecewise linear slopes from $\eta_u$ to $\eta_e$ and $\eta_e$ to $\eta_d$. The open-ended nature of the $\eta(x)$ discretization should allow future development of a variety of finite-volume forms that can be easily demonstrated to be well-balanced and consistent with the above derivations.

By its definition in eq. (37) with constraint eq. (38), the weighting function $\lambda(x)$ is an abstraction of the topographic pressure distribution over a finite volume, which is affected by both topography and the local slope of the free surface, as illustrated in Figs. 3, 5, and 6. The $\lambda(x)$ is more complicated and abstract than $\eta(x)$ because the free-surface elevation is approximated

as uniform across the channel at any $x$ location, but $\lambda(x)$ represents the integrated effect of complex 3D topography, e.g. the

10 $A_{R(m)}$ stair-step as illustrated in Fig. 4. The key point is that the $\lambda(x)$ weighting function has an integral constraint $\int \lambda \, dL = 1$ because the change in the cross-sectional area over the control-volume flux faces, $A_d - A_u$, has already been extracted from the integral of $PA_R$ through eq. (39). A comparison of Figs. 5 and 6 shows that $\lambda(x)$ is *not* independent of $\eta(x)$ and hence arguably should be a moderator between the static topography and the dynamics of the free surface. However, development of $\lambda(x,t)$ forms that are dynamically dependent on $\eta(x,t)$ are beyond the scope of the present work. Herein, we appeal to

15 Occam's razor and note that the simplest $\lambda(x)$ that satisfies the integral constraint is $\lambda(x) = L_e^{-1}$, where $L_e$ is the control volume length, illustrated in Fig. 1. This choice of $\lambda(x)$ is a zero-order polynomial implying the stair-steps are uniformly distributed over the element, as illustrated in Fig. 3. Clearly, for the nonlinear cases in Figs. 5 and 6 this form of $\lambda(x)$ is an approximation that reduces nonlinear interaction between the free surface and the topography. Arguably, this is consistent with a discrete scheme using a single-valued geometry function such as $\eta_e = f(V_e)$ that neglects the effect of the free surface slope on the relationship between volume and surface elevation. Note that because the $A_d - A_u$ area is already extracted from the integral, using $\lambda(x) = L_e^{-1}$ ensures that the $T_e$ term is exact at the linear limit and a bounded approximation for nonlinear interactions. That is, with $\lambda(x) = L_e^{-1}$ it follows that

$$T_e = \frac{g\,(A_d - A_u)}{L_e} \int_L \eta(x)\,dx \quad \leq \quad g\,(A_d - A_u)\max\left(\eta(x)\right)$$

Thus, the largest possible value for the $T_e$ term is based on the maximum piezometric pressure over the control volume acting on the difference between upstream and downstream areas.

The simple form of $\lambda(x) = L^{-1}$ is consistent with either increasing cross-sectional area ($\gamma(x) > 0$ over $L_e$) or decreasing cross-sectional area ($\gamma(x) < 0$ over $L_e$), but is questionable for a non-monotonic case (e.g., Fig. 6). To examine this issue in more detail, consider the somewhat simpler case of a linear free surface slope with cross-sectional area that increases along

the streamwise direction in the upper section and decreases in the lower, as shown in Fig. 7. The $\lambda(x) = L^{-1}$ approach cannot represent the actual change in cross-section area, but instead provides a linear trend between $A_u$ and $A_d$ as shown in Fig. 8. In contrast, if the control volume in Fig. 7 were split into two separate volumes at the centerline (i.e., the gray dashed line), then the stair-steps of Fig. 7 would be readily represented by the $\lambda(x) = L^{-1}$ approximation as both control volumes would be monotonic.

Comprehensive investigation of different forms for $\eta(x)$ and $\lambda(x)$ in the $T_e$ term is clearly needed and will take significant future effort. For the present purposes, we examine the simplest polynomial forms for $\eta(x)$ in combination with the zeroth-order $\lambda(x) = L^{-1}$. To provide insight into how a higher-order $\lambda(x)$ discretization adds complexity in the derivation, we can derive a simple linear form of $\lambda(x)$ that depends only on topography and couple with a linear form of $\eta(x)$. Such a $\lambda(x)$ is unlikely to be useful with a dynamic free surface, but is illustrative of the complexity that can be developed with quadrature

of even simple linear equations. We will use the nomenclature $T_{e(m,n)}$ to designate an $m$-order polynomial for $\eta(x)$ and an $n$-order polynomial for $\lambda(x)$.

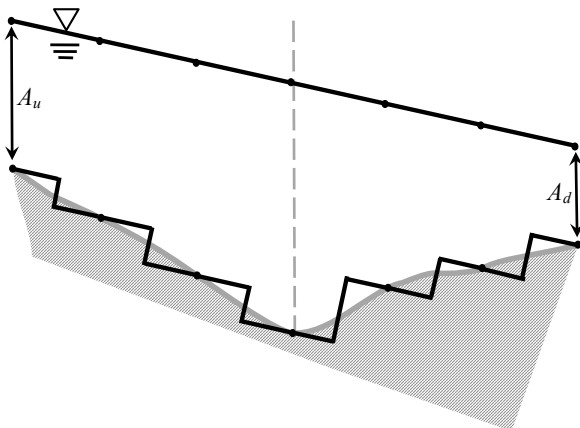

**Figure 7.** Non-monotonic stair-steps of cross-sectional area with linear monotonic free surface. Gray dashed line is where the cross-section area reverses from increasing to decreasing.

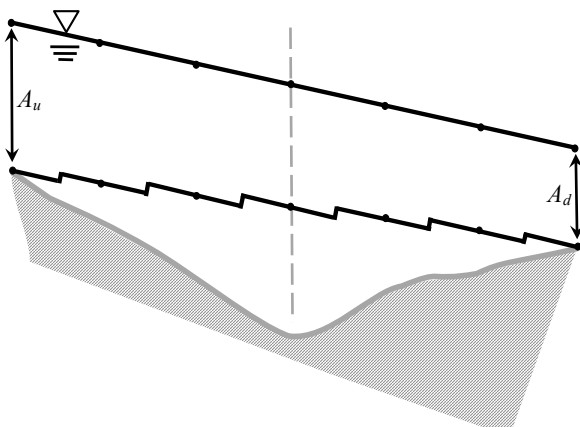

**Figure 8.** Approximated finite-element cross-section characteristics using $\lambda(x) = L_e^{-1}$ for non-monotonic topography in Fig. 7. Gray dashed line is where the cross-section area reverses from increasing to decreasing.

**The $T_{e(0,0)}$ approximation**

The simplest approximations arise by assuming uniform values such that $\eta(x) \approx \bar{\eta}$ and $\lambda(x) \approx L_e^{-1}$, where $\bar{\eta}$ is the average water surface elevation over the element and we recall that $\int_L \lambda \, dx = 1$. It follows that

$$T_{e(0,0)} \equiv g\left(A_d - A_u\right)\bar{\eta} \tag{50}$$

If we let $\eta_e \approx \bar{\eta}$, then eq. (48) can be written as

$$L_e \frac{\partial Q_e}{\partial t} - \frac{Q_u^2}{A_u} + \frac{Q_d^2}{A_u} - gA_u\eta_u + gA_d\eta_d = g\left(A_d - A_u\right)\eta_e - gV_e S_{f(e)} \tag{51}$$

Note that the RHS in the above is what we might infer as a discrete finite-volume version of $g\eta \, \partial A / \partial x$ in the differential source term of eq. (47).

**The $T_{e(1,0)}$ approximation**

If we represent the free surface by a 1st-order polynomial, $\eta(x) = ax + b$, while retaining the zeroth-order $\lambda(x) = L_e^{-1}$ we obtain

$$T_{e(1,0)} = g\left(A_d - A_u\right)\left(\frac{aL}{2} + b\right) \tag{52}$$

A linear approximation is consistent with a free surface where $b = \eta_u$ and $a = -\left(\eta_u - \eta_d\right)/L$, so we obtain a finite-volume source term

$$T_{e(1,0)} = \frac{1}{2}g\left(A_d - A_u\right)\left(\eta_u + \eta_d\right) \tag{53}$$

Note that the above implies products of $A_d\eta_d$ and $A_u\eta_u$ in the source term that can be moved to the LHS as part of the conservative flux terms. Substituting $T_{e(1,0)}$ for $T_e$ in eq. (48) and redistributing terms provides

$$L_e \frac{\partial Q_e}{\partial t} - \frac{Q_u^2}{A_u} + \frac{Q_d^2}{A_u} - \frac{1}{2}gA_u\eta_u + \frac{1}{2}gA_d\eta_d = \frac{1}{2}g\left(A_d\eta_u - A_u\eta_d\right) - gV_e S_{f(e)} \tag{54}$$

Thus, the model of a linear free surface and uniform $\lambda(x)$ serves to change the weighting of the $gA\eta$ terms (from unity to $1/2$) and provides a source term that contains only face values of $A$ and $\eta$, which is unlike eq. (51) that requires an element approximation of $\eta_e$. Interestingly, the above finite-volume form does *not* have a differential representation as $L \to 0$. That is, the free-surface differential source term in eq. (47) is based on $(A_d - A_u)/L \to dA/dx$ as $L \to 0$. However, once we have chosen relationships for $\eta(x)$ and $\lambda(x)$ and moved a portion of the source term into the fluxes, an attempt to create a differential out of our source term encounters the form

$$\lim_{L \to 0} \frac{A_d\eta_u - A_u\eta_d}{L_e}$$

which can only be reduced to a differential by making additional approximations as to the behavior of $A$ and $\eta$ at the faces of the finite volume. As a further insight, for the special case of a rectangular channel with frictionless flow eq. (54) can be

transformed by using $A = BH$ where $B$ is the breadth and $H$ is the depth, so that we obtain a finite volume form of

$$L_e \frac{\partial}{\partial t} (H_e U_e) - H_u U_u^2 + H_d U_d^2 + \frac{1}{2} g \left( H_d^2 - H_u^2 \right) = g \frac{(H_d + H_u)}{2} (z_u - z_d) \tag{55}$$

As $L_e \to 0$, and $S_0 = -\partial z / \partial x$, the above implies a conservative differential form of

$$\frac{\partial HU}{\partial t} + \frac{\partial}{\partial x} \left( HU^2 \right) + \frac{1}{2} g \frac{\partial H^2}{\partial x} = g H S_0 \tag{56}$$

which is commonly used in studies of simplified conservative forms (e.g., Bouchut et al., 2003; Hsu and Yeh, 2002). Thus the $T_{e(1,0)}$ is also consistent with prior differential forms.

**The $T_{e(2,0)}$ approximation**

We can take this approach further by approximating the free surface as a parabola based on $\{\eta_u, \eta_e, \eta_d\}$ where $\eta_e$ is a characteristic free surface height at the center of the finite volume. Using $\eta(x) = ax^2 + bx + c$ with $x = 0$ at $A_u$ and $x = L$ at $A_d$, provides

$$c = \eta_u \tag{57}$$

$$b = \frac{1}{L} \left( -3\eta_u + 4\eta_e - \eta_d \right) \tag{58}$$

$$a = \frac{2}{L^2} \left( \eta_u - 2\eta_e + \eta_d \right) \tag{59}$$

Using $\lambda(x) = L^{-1}$, results in

$$T_{e(2,0)} = \frac{g}{6} \left( A_d - A_u \right) \left( \eta_u + 4\eta_e + \eta_d \right) \tag{60}$$

It is useful to multiply through and regroup terms so that $A_d \eta_d$ and $A_u \eta_u$ are isolated. Where these terms are balanced, they
can be moved to the LHS as conservative terms. Regrouping provides:

$$T_{e(2,0)} = \frac{g}{6} \left\{ A_d \eta_d - A_u \eta_u + A_d \left( \eta_u + 4\eta_e \right) - A_u \left( 4\eta_e + \eta_d \right) \right\} \tag{61}$$

So our momentum equation can be written as

$$L_e \frac{\partial}{\partial t} (Q_e) - \frac{Q_u^2}{A_u} + \frac{Q_d^2}{A_d} - \frac{5}{6} g A_u \eta_u + \frac{5}{6} g A_d \eta_d = \frac{1}{6} g \left\{ A_d \left( \eta_u + 4\eta_e \right) - A_u \left( 4\eta_e + \eta_d \right) \right\} - g V_e S_{f(e)} \tag{62}$$

Once again, the specific representation of $\eta(x)\lambda(x)$ provides a modification of the coefficient of the $gA\eta$ terms in the conservative fluxes and sets the form of the non-conservative source term. This form does not appear to readily reduce to any differential form that is previously seen in the literature, and thus provides an interesting new avenue for investigation.

## The $T_{e(1,1)}$ approximation

The above forms used a uniform $\lambda(x) = L^{-1}$. We can readily extend the concept to analytical forms of $\lambda(x)$, although it is not clear that such increasing complexity will yield an advantage in the design of a numerical model. The $\lambda(x)$ function is a weighting function that reflects distribution of the bottom elevation stair-steps, as described in §4. The only restriction on $\lambda(x)$ is that it must integrate to unity over $L_e$. Physically, as illustrated in Figs. 3, 5, and 6, the $\lambda(x)$ should be a function of $\eta(x)$ as well as the topography. However, we do not (as yet) have a good working framework for a dynamic representation of $\lambda(x, \eta)$. Thus, to illustrate the complexities that arise with a non-uniform $\lambda(x)$, herein we will simply analyze a somewhat arbitrary static linear relationship where $\lambda(x) = \alpha z(x)$, where $z$ is the bottom elevation and $\alpha$ is a scaling constant to ensure $\int_L \lambda \, dx = 1$. We introduce linear approximations of the bottom as $z(x) = Ax + B$ and the free surface as $\eta(x) = ax + b$. We can write these approximations as

$$z(x) = -\frac{1}{L}(z_u - z_d)x + z_u \tag{63}$$

$$\eta(x) = -\frac{1}{L}(\eta_u - \eta_d)x + \eta_u \tag{64}$$

Using $\int_L \alpha z(x) = 1$, it can be shown that

$$\alpha = \frac{2}{L(z_u + z_d)} \tag{65}$$

Using eq. (49), a quadrature problem can be presented as

$$\frac{T_{e(1,1)}}{g(A_d - A_u)} = \alpha \int_L z(x)\eta(x) \, dx$$

$$= \frac{2}{L(z_u + z_d)} \int_L \left[ -\frac{1}{L}(z_u - z_d)x + z_u \right] \left[ -\frac{1}{L}(\eta_u - \eta_d)x + \eta_u \right] dx \tag{66}$$

After some algebra, we find

$$\frac{T_{e(1,1)}}{g(A_d - A_u)} = \frac{2\eta_u z_u + \eta_u z_d + \eta_d z_u + 2\eta_d z_d}{3(z_u + z_d)} \tag{67}$$

Unfortunately, for the above form we cannot use the redistribution trick to split $T_e$ and move a portion to the LHS of eq. (48). The problem is that any split term will have $z_u + z_d$ in the denominator, which will not be conservative when used as a coefficient on a control volume face. Thus, the $T_{e(1,1)}$ form of momentum is

$$L_e \frac{\partial Q_e}{\partial t} - \frac{Q_u^2}{A_u} + \frac{Q_d^2}{A_u} - gA_u\eta_u + gA_d\eta_d = g(A_d - A_u)\left(\frac{2\eta_u z_u + \eta_u z_d + \eta_d z_u + 2\eta_d z_d}{3(z_u + z_d)}\right) - gA_e S_{f(e)} \tag{68}$$

In general, it appears that only the uniform $\lambda(x) = L_e^{-1}$ form will allow us to shift of part of the source term onto the flux side. Any form that has a non-uniform $\lambda(x)$ must necessarily have a dependency on $z(x)$ in the cell to satisfy $\int_L \lambda \, dx = 1$. We cannot create a conservative term at a face when the value computed from the downstream cell depends on a value within the downstream volume and while the value computed from the upstream cell depends on a value within the upstream volume.

Note that this particular $T_{e(1,1)}$ form is for demonstration purposes and is *not* recommended for use in any numerical scheme. This $T_{e(1,1)}$ is predicated on an assumed weighting of $\lambda(x) = \alpha z(x)$, which does not have a physical linkage to specific cross-section geometry or expected flow conditions.

## 10  Summary of approximate forms

The $T_{e(0,0)}, T_{e(1,0)}, T_{e(2,0)}$ and $T_{e(1,1)}$ approximations all follow a similar form

$$L_e \frac{\partial Q_e}{\partial t} - \frac{Q_u^2}{A_u} + \frac{Q_d^2}{A_d} - gA_u\eta_u(1 - \delta_u) + gA_d\eta_d(1 - \delta_d) = K_{e(m,n)} - gV_eS_{f(e)} \tag{69}$$

where $0 \le \delta_u, \delta_d \le 1$ are fixed coefficients and $K_{e(m,n)}$ is a time-space-varying topographic source term, whose exact forms are determined by the approximations used for $T_e$. This form was suggested in Section 2 with eq. (9) based on a philosophical argument of moving as much as possible of the RHS source terms to the conservative LHS flux terms. The key point for future work is that these forms (with the exception of $K_{e(1,1)}$) are relatively straightforward in their representation of values that are the natural elements of a SVE computational model for river networks and urban drainage. This approach eliminates the need for estimating or computing the $I_1$ and $I_2$ of the Cunge-Liggett conservative form and replaces them with the simple cross-sectional area term and a $K_e$ that is computed from discrete values of $\eta$, and $A$. Values for $\delta_u$, $\delta_d$ and $K_e$ for these forms are presented in Table 2. Examination of the above leads to the conclusion that the use of any polynomial representation of $\eta(x)$ with $\lambda(x) = L^{-1}$ will produce a $K_{e(m,0)}$ source term that will exactly vanish when the free surface is flat, e.g., $\eta_u = \eta_e = \eta_d$. Thus, these schemes are inherently "well-balanced" as discussed in Section 3. Furthermore, for these cases the source terms will be Lipschitz smooth as long as the solution variables are smooth.

## 6  Summary and Discussion

The conservative differential form of the non-hydrostatic version of the Saint-Venant equations, simplified from the derivation in Section 4, can be written as

$$\frac{\partial AU}{\partial t} + \frac{\partial}{\partial x}\left(\left[\beta AU^2 + gA\eta + \frac{\breve{P}}{\rho}\right]\cos\psi\right) = g\eta\frac{\partial A}{\partial x}\cos\psi + \frac{1}{\rho}\int_{A_R}\breve{P}(z)dA - gAS_f + m_e \tag{70}$$

where $m_e$ is the source/sink of momentum from lateral fluxes per unit length (i.e., $M_eL^{-1}$). This equation is similar to previous work but includes both non-hydrostatic terms and effects of free surface slope ($\cos\psi$) that are often neglected. The key contribution of the present work is the semi-discrete, conservative, finite-volume form that corresponds to the differential form above:

$$\frac{\partial}{\partial t}(U_eV_e) - \beta_u A_u U_u^2 \cos\psi_u + \beta_d A_d U_d^2 \cos\psi_d$$

$$- gA_u\eta_u\cos\psi_u + gA_d\eta_d\cos\psi_d - \frac{\cos\psi_u}{\rho}A_u\breve{P}_u + \frac{\cos\psi_d}{\rho}A_d\breve{P}_d$$

$$= g(A_d\cos\psi_d - A_u\cos\psi_u)\int_L \eta(x)\lambda(x)\,dx + \frac{1}{\rho}\int_L \gamma(x)\int_{A_R}\breve{P}(x,y,z)\,dA\,dx - gV_eS_{f(e)} + M_e \tag{71}$$

**Table 2.** Values for conservative $A\eta$ coefficients and $K_{e(m,n)}$ source terms for finite volume schemes with $(m,n)$ polynomial approximations of $T_e$. Note that forms using $(m,n)=(m,1)$ are not recommended without further theoretical development and are provided only for illustrative purposes.

| $(m,n)$ | $\delta_{u(m,n)}$ | $\delta_{d(m,n)}$ | $K_{e(m,n)}$ |
|---|---|---|---|
| $(0,0)$ | $0$ | $0$ | $g\left(A_d - A_e\right)\eta_e$ |
| $(1,0)$ | $1/2$ | $1/2$ | $g\left(A_d\eta_u - A_u\eta_d\right)/2$ |
| $(2,0)$ | $1/6$ | $1/6$ | $g\left\{A_d\left(\eta_u + 4\eta_e\right) - A_u\left(\eta_d + 4\eta_e\right)\right\}/6$ |
| $(1,1)$ | $0$ | $0$ | $g\left(A_d - A_u\right)\left(\frac{2\eta_u z_u + \eta_u z_d + \eta_d z_u + 2\eta_d z_d}{3(z_u+z_d)}\right)$ |

where $\breve{P}_u$ and $\breve{P}_d$ are the average non-hydrostatic pressures on the upstream and downstream cross-sectional areas. In this finite-volume form, the only approximations introduced are: (1) uniform-density incompressibility, (2) the effects of momentum redirection around bends is either negligible or is handled in friction terms, (3) the cross-channel variability in the free-surface slope is negligible, and (4) a friction slope model can be used to represented integrated viscous effects. In addition we have introduced a geometric restriction that the upstream and downstream cross-sections must be vertical planes that are orthogonal (in the horizontal plane) to the mean flow direction. The above form can be used to analyze systems that include non-hydrostatic pressure and slope gradients beyond the small-slope approximation of the traditional SVE. As warning for future development of non-hydrostatic methods, note that the fundamental 1D derivation is effectively treating the non-hydrostatic pressure gradients in the horizontal as absorbing and redirecting momentum along the curving channel. Thus a part of the this term is, in effect, encapsulated within the approximation that allows $A_u$ and $A_d$ to be cross-sections that are not parallel.

The principal feature of the new finite-volume formulation is the topographic source term $\int \eta(x)\lambda(x)\,dx$ that can be represented by analytical functions to approximate a smoothly-varying free surface and its interaction with topography across the finite-volume element. Discrete polynomial representations of $\int \eta(x)\lambda(x)\,dx$ have been evaluated in Section 5, with the resulting topographic terms designated as $T_{e(m,n)}$ for an $m$-degree polynomial representing $\eta$ and an $n$-degree polynomial representing $\lambda$. In an approximate form, the $T_{e(m,n)}$ term is split into a $\delta_{u(m,n)}$ factor applied to $gA_u\eta_u$ and a $\delta_{d(m,n)}$ factor

applied to $gA_d\eta_d$, which become part of the conservative flux terms. The remainder of the $T_{e(m,n)}$ becomes a $K_{e(m,n)}$ source term in the approximate finite-volume form. Simpler conservative finite-volume forms that use the common approximations of the hydrostatic equations ($\breve{P} \approx 0$) with small free surface slope ($\cos\psi \approx 1$), uniform cross-sectional velocity ($\beta \approx 1$), no

momentum sources ($M_e = 0$), can be written as

$$L_e \frac{\partial Q_e}{\partial t} - \frac{Q_u^2}{A_u} + \frac{Q_d^2}{A_d} - gA_u\eta_u\left(1 - \delta_{u(m,n)}\right) + gA_d\eta_d\left(1 - \delta_{d(m,n)}\right) = K_{e(m,n)} - gL_eA_eS_{f(e)} \tag{72}$$

where the discrete $K_e$ and $\delta$ terms are shown in Table 2. It is worthwhile to compare the above to a finite-volume form derived using the Cunge-Liggett form of the SVE, which could be written (using the $I_1$ and $I_2$ definitions) as

$$L_e \frac{\partial Q}{\partial t} - \frac{Q_u^2}{A_u} + \frac{Q_d^2}{A_d} - g\int_{H(u)} (H-z)B(z)\,dz + g\int_{H(d)} (H-z)B(z)\,dz = g\int_L\int_H (H-z)\frac{\partial B}{\partial x}\,dx + g\int_L A(S_0 - S_f)\,dx \tag{73}$$

Performance of the traditional scheme depends on the specification for $B(x,z)$ that defines the irregular bathymetry of the channel. Although the $B(x,z)$ term is developed without any approximations, it is a non-trivial matter to simplify these terms to create practical computational forms for irregular cross-section geometry. The source term on the RHS of the Cunge-Liggett form is effectively an integration of both variations in the channel topography and the water surface elevation over the volume – similar to the new $T_e$ – but without a limiting constraint (i.e., $\partial B/\partial x$ is not inherently limited in magnitude for irregular

topography). Furthermore, the selection of $B(x,z)$ for the RHS affects the integration on the LHS hydrostatic pressure term – thus obtaining a "well-balanced" source term that compensates for $S_0$ will also affect the conservative flux terms. The $\lambda(x)$ used to develop the $K_e$ term eq. (72) serves a similar purpose to the source-term integral in the Cunge-Liggett form, but provides a simple weighting function that can be analytically integrated with an approximation for $\eta(x)$ and is inherently constrained such that $\int \lambda\,dx = 1$ over a control volume. The other major difference between the two approaches is that eq. (73) uses $S_0$ as

a source term on the RHS of the equations whereas in the new approach eq. (72) dispenses with this artifice so that the source terms only include friction (which is guaranteed to damp momentum) and the portion of the topographic effects that cannot be transferred into the conservative $\delta_u$ and $\delta_d$ terms.

There is a long history of using of $S_0$ in the source term in the SVE, and it is indeed part of the author's prior model (Liu and Hodges, 2014). However, use of $S_0$ with irregular geometry brings the problems of creating a "well-balanced" conservative

scheme, as discussed in Section 3. Furthermore, the use of $S_0$ in the source term requires the pressure term to be treated as the hydrostatic pressure rather than the piezometric pressure, as shown in Section 4. Because the hydrostatic pressure is a function of depth, its integration over a cross-sectional area requires knowledge of the distribution of depth across the channel – a significant computational complexity for irregular topography. In contrast, the integration of the piezometric pressure over a cross-section is exactly $PA$ and does not require knowledge of how depth is distributed across the channel. Other authors have noted similar problems: Schippa and Pavan (2008) derived a conservative differential form that retained $gI_1$ in the flux terms and removed $S_0$ by showing that it could be combined with the $I_2$ source term as $\partial I_1/\partial x$ for a uniform water level. It seems that the Schippa and Pavan (2008) differential equation might be preferred to the approach proposed herein for high-resolution topography with small grid spacing (i.e., where we have confidence that the computation of $I_1$ is meaningful). However, at

larger scales where geometric cross-sections are broadly spaced and the computation of $I_1$ is questionable, the simplicity of

using $A$ and $\eta$ for piezometric pressure gradient terms is likely to be preferred. Other authors, notably Rosatti et al. (2011), have simply accepted $gA\partial\eta/\partial x$ as an unavoidable source term rather than dealing with the problems of obtaining a well-balanced method with the hydrostatic pressure.

Since $S_0$ was not in Alexandre de Saint-Venant's original paper, how did it come to be commonly used in the Saint-Venant equations? Arguably there are two sources associated with different simulation scales: (1) in hydrology the kinematic wave model provides $S_0 = S_f$, which leads to prioritization of $S_0$ as a hydraulic parameter, and (2) in mathematics the equation:

$$\frac{\partial hu}{\partial t} + \frac{\partial hu}{\partial x} = s \tag{74}$$

is the canonical form of a 1D inhomogenous hyperbolic advection equation for depth $h$ and velocity $u$, which leads to a prioritization of the depth as a fundamental parameter and requires $S_0(x)$ be relegated to the source term for irregular geometry. However, for solution of the SVE there is no need to exactly mimic the hyperbolic advection equation to obtain a conservative form, thus there is no need to introduce $S_0$. The above comparisons of the Cunge-Liggett form and the new finite-volume form illustrate the additional complexity of introducing $S_0$. Beyond these issues is a more fundamental problem: numerical methods for inhomogeneous partial differential equations are only well-posed if the source terms are Lipschitz smooth (e.g. Iserles, 1996) – otherwise one should not be surprised by numerical instabilities and/or difficulties in convergence. Any river model that uses raw data from surveyed cross-sections will inherently have non-smooth $S_0(x)$. As a result, much of the computational complexity is likely to be compensating for the lack of Lipschitz smoothness in the boundary conditions of topography. In contrast, when the free-surface elevation (piezometric pressure) is used instead of depth (hydrostatic pressure), then $S_0(x)$ disappears from the SVE and the smoothness of the source term is assured (for smooth solution variables) by the choice of smooth functions for $\lambda(x)$ and $\eta(x)$ and the friction slope model $S_f(x)$. In general, as long as the solutions of $\eta$ and $Q$ remain smooth the source term of the SVE, as derived herein, should remain smooth. That is, the approach herein cannot guaranty a smooth solution, but it can guaranty that any observed non-smoothness in the source terms during a simulation is a result of non-smoothness in the solution variables rather than direct forcing of boundary conditions.

As a matter of pure speculation, the new form of the finite-volume equations brings up some interesting possibilities for large-scale modeling. Imagine that we would like to model a river network or urban drainage network where we have some high-resolution (1×1 m) data in some areas but not in others. Let us also say our computer power limits us to a SVE solution with a median cell length of about 20 m. Can we use our high resolution knowledge directly at the coarse scale? Arguably, the $\lambda(x)$ approach could give us a means of directly incorporating effects of subgrid-scale topography into a single source term – however, significant theoretical work is required to develop a consistent methodology that retains the fundamental "well-balanced" and Lipschitz smooth characteristics of the finite-volume equations.

Clearly, there remains much practical experimentation to be done in comparing various forms of $T_{e(m,n)}$ and the effectiveness of different numerical solution methods with different $\eta(x)$ polynomial representations. There is also a need to develop a firm theoretical relationship between $\lambda(x)$ to $\eta(x)$ to overcome the difficulties illustrated with Figs. 5 and 6 where nonlinear interactions between the free surface and topography cannot be readily represented with the uniform $\lambda(x) = L^{-1}$ applied herein. Finally, there are substantial questions on whether the new approach can be used with methods designed to satisfy the

entropy criterion (e.g. Greenberg and Leroux, 1996; Harten et al., 1983; Lax, 1973). It is hoped that researchers will consider adapting their finite-volume codes to the form of eq. (72) and reporting their experience.

## 7    Conclusions

New finite-volume forms of integral momentum equations for unsteady flow in open channels with varying cross-sections have been derived in this paper. These equations reduce to the classic differential forms of the Saint-Venant equations (under the $T_{e(0,0)}$ and $T_{e(1,0)}$ approximations) and also provide new approximate finite-volume forms that are suitable for analytical representations of topography and free surface elevation over a finite-volume element. The new forms use the piezometric pressure (free-surface elevation) rather than hydrostatic pressure (depth) as a fundamental variable, and thus do *not* include the

channel slope, $S_0$. This approach provides a cleaner finite-volume form as the nonlinear interactions of topography and the free surface are handled in a single integral term, $g\left(A_d - A_u\right)\int \eta\lambda\,dx$ where $\lambda$ is a quadrature weighting function and $A_d - A_u$ is the downstream increase in cross-sectional area of the control volume. The introduction of $\lambda(x)$ provides a potential avenue to convert practical knowledge of fluid/geometry variations within a control volume into source and flux terms of the finite-volume equations. The derivations herein can be used to generate a variety of conservative, well-balanced, finite-volume forms for the

SVE that could be employed with a wide range of numerical discretization schemes. This work provides only the theoretical development for the new finite-volume equations; the practical implementation and numerical testing remains a subject for future work.

*Code availability.* A demonstration code for discretized application of the new SVE form is available on GitHub at https://github.com/benrhodges/SvePy. This code is associated with a companion paper Hodges and Liu (2018) that explains the discretized form.

**Appendix A**

To elaborate on eq. (19), the advection of some quantity $\phi$ through a control volume bounded by surface $S$ can be represented as

$$\oint_S \phi\, u_k n_k \, dA = \oint_S \phi\, u_{\hat{n}} \, dA \tag{A1}$$

where $u_{\hat{n}}$ is the local normal velocity across a flux surface. For a flux surface of area $A$, we define the nonlinearity of distribution

across the surface as

$$\alpha \equiv \frac{1}{A\Phi U_{\hat{n}}} \int_A \phi\, u_{\hat{n}} \, dA \tag{A2}$$

where $\Phi$ is the average of $\phi$ over flux area $A$, and $U_{\hat{n}}$ is the average surface normal velocity. Note that because $u_{\hat{n}}$ and $U_{\hat{n}}$ are derived from projection of the velocity onto the surface normal vector, they have positive signs for outflows and negative

sign for inflows. Consistency of $u_{\hat{n}}$ and $U_{\hat{n}}$ ensures that $\alpha \geq 0$; however, this introduces a nomenclature difficulty because the sign does not depend solely on the nominal upstream or downstream direction of the flow. To simplify the nomenclature, we consider only upstream ($u$) and downstream ($d$) flux surfaces for a control volume, and represent the normal velocity by $U_{\perp}$ such that a downstream velocity on any face is $U_{\perp} > 0$ and an upstream velocity on any face is $U_{\perp} < 0$. It follows that

$$\oint_S \phi u_k n_k \, dA = -\left(\alpha \Phi U_{\perp} A\right)_u + \left(\alpha \Phi U_{\perp} A\right)_d \tag{A3}$$

where

$$\alpha \equiv \frac{1}{A \Phi U_{\perp}} \int_A \phi u_{\perp} \, dA \tag{A4}$$

Let $Q \equiv U_{\perp} A$ with the same sign conventions as $U_{\perp}$, so that

$$\oint_S \phi u_k n_k \, dA = -\left(\alpha \Phi Q\right)_u + \left(\alpha \Phi Q\right)_d \tag{A5}$$

Consider a channel with a linear free surface so that a velocity parallel to the free surface ($u_{\hat{i}}$) has the same direction over the entire control volume. Let $\phi = u_{\hat{i}}$ and $\Phi = U_{\hat{i}}$ so we obtain

$$\oint_S u_{\hat{i}} u_k n_k \, dA = -\left(\alpha U_{\hat{i}} Q\right)_u + \left(\alpha U_{\hat{i}} Q\right)_d \tag{A6}$$

where $U_{\hat{i}}$ is interpreted as the average velocity parallel to the free surface with the same sign conventions as $U_{\perp}$. For a free surface at angle $\psi$ from the horizontal it follows that

$$\cos \psi = \frac{U_{\perp}}{U_{\hat{i}}} \tag{A7}$$

Using eq. (A4) with $\Phi = U_{\hat{i}}$ and $U_{\perp} = U_{\hat{i}} \cos \psi$ provides

$$\alpha = \frac{1}{A \left[U_{\hat{i}}\right]^2 \cos \psi} \int_A u_{\hat{i}} u_{\perp} \, dA \tag{A8}$$

Noting that $u_{\hat{i}} = u_{\perp} \cos^{-1} \psi$ we obtain

$$\alpha = \frac{1}{A \left[U_{\hat{i}}\right]^2} \int [u_{\hat{i}}]^2 \, dA \tag{A9}$$

Thus, $\alpha$ of eq. (A4) for $\Phi = U_{\hat{i}}$ is identical to $\beta$ of eq. (18). It follows that

$$\oint_S u_{\hat{i}} u_k n_k \, dA = -\left(\beta U_{\hat{i}} Q\right)_u + \left(\beta U_{\hat{i}} Q\right)_d \tag{A10}$$

which is eq. (19). However, because $Q \equiv U_{\perp} A$ it follows that $Q = U_{\hat{i}} A \cos \psi$. Thus, equivalent forms are

$$\oint_S u_{\hat{i}} u_k n_k \, dA = -\left(\beta U_{\hat{i}}^2 A \cos \psi\right)_u + \left(\beta U_{\hat{i}}^2 A \cos \psi\right)_d \tag{A11}$$

$$= -\left(\beta \frac{Q^2}{A \cos \psi}\right)_u + \left(\beta \frac{Q^2}{A \cos \psi}\right)_d \tag{A12}$$

which all reduce to conventional forms with $Q = UA$ when $\cos \psi = 1$.

*Competing interests.*   The author declares that he has no conflicts of interest.

     *Acknowledgements.*   This article was developed under Cooperative Agreement No. 83595001 awarded by the U.S. Environmental Protection
     Agency to The University of Texas at Austin. It has not been formally reviewed by EPA. The views expressed in this document are solely
     those of the author and do not necessarily reflect those of the Agency. EPA does not endorse any products or commercial services mentioned
     in this publication. The author appreciates the helpful discussion with the reviewers that pointed out places where the draft manuscript was
unclear in presenting the conceptual model.

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
