# Peer review of "Conservative finite-volume forms of the Saint-Venant equations for hydrology and urban drainage"

_Hydrology and Earth System Sciences, 2018_

## Referee Comment (RC1) · Anonymous Referee #1 · 10 Sep 2018

The paper presents new conservative forms of Saint Venant Equations. It reviews classical forms of SV equations and introduces new ways to present it. According to the manuscript, these new SV eq may be more appropriated for applications in hydrology and urban drainage. Saint Venant equations are one of the most important in water resources field but it is still not always easy to solve it in hydrodynamic models due to the nature of the problem. Improving methods for hydrodynamic modelling is an important challenge, given new input that data types and availability, and applications. So the goal of the manuscript is relevant. It is generally well written and the ideas proposed seem correct. However, I fell that the paper would benefit from a few improvements to make it an stronger contribution: (i) better intro based on previous work (ii) better

explanations on the equations, with more figures showing concepts (iii) a few examples showing the value of the proposed new forms of SV equations. Detailed comments are provided below. I hope it can help the author to improve the manuscript.

Introduction:

The intro section focuses on detailed explanations of different forms of SV equations that helps to identify how to write SV model in a more conservative form. The manuscript argues that this new form is proposed to address challenges in modelling large scale river networks and urban drainage system. However, there is no clear demonstration of past research showing that the major modelling challenges are associated with the form of SV equations. For example, there are several recent research using the local inertia approximation of SV solved with an explicit scheme proposed by Bates et al. (2010), that is now used in several models (Camaflood, Yamazaki et al., 2013; Lisflood-FP, Neal et al., 2012; MGB, Pontes et al., 2017, Sigueira et al., 2018) to simulate large basins and continental to global scale. But this past research did not documented problems related to forms of SV equation. What are the current difficulties? What could be the benefits of a new one ? Improved model stability? Accuracy? For example the scheme from Bates 2010 neglects advective inertia. ... computational efficiency? These issues should be discussed in the introduction, based on previous research, to convince a broader audience that this new form of SV equations could be useful to improve their simulations. Also, most of the detailed explanations presented in the introduciton should be moved to section 2.

Page 3 Line 20 - 25. Equation 8. Explain why f1 and f2 are necessary in eq 8 while it was not necessary in eq. 7.

Section 2. Page 4 line 7. Review the frictional force per unit area.

Page 4 lines 17 – 29. Also review recent work using local inertia aproximation from Bates et al. 2010 for large scale river networks.
Page 4 lines 17 – 29. Review work for short scale using HEC-RAS.

Page 5 Lines 5-15. Please discuss explicit vs implicit sechemes. Discuss local inertia explicit formulation. Review 6 point Abbot and Ionescu scheme (1967). Explain that Preissman scheme is a finite difference based on integral relations that improve conservation.

Section 3:

Please use more figures to help the reader to understand the paper.

Equation 13. Please use a figure to define the control volume Page 8 line 2. "This vector is local and change along the channel ....". Please use a figure to clarify it.

Page 8 line 25. "It is known that ....". Please cite references showing these arguments. Page 9 Line 13. Figure defining the free surface slope n(x), these angles, etc. Eq 25. Why P(x) is independent to z? How the shape of cross section and its variability is considered?

Page 12 equations 27, 28, 29. What is m? What lambda represents? Is it just a mathematical trick? Or does it have any physical meaning? How it relates to zb(x)? How it relates to river cross section shape and its variation along x? Why int (lambda) = 1? (eq 28)

Equation 32. Ve is confusing here. It is volume but one can make a confusion with flow velocity. Please, use a good figure to define the control volume, what is Ve, Le, upstream and downstream cross section, etc.etc...

Equation 33. How gAn was obtained? Define how gAn is derived from P, using water surface n instead of depth H.

Section 4. Please define the meaning of lambda function. It is important to understand the choice for polynomial approximations.

Please show in a figure the approximations T(0,0), T(1,0), T(2,0), T(1,1) of n(x) and
lambda(x) or zb(x).

Equation 60. Check it. I guess the last term is g.Ae.Le.Sfe.

Pg 19, line 22. "S0 brings a host of problems". What problems? Please show it based on past publications.

Eq. 62. Not clear.

Final discussions and conclusions. In my view, the paper would be much more convincing if a few examples showing that application of the new forms of SV equations provide the same results as other classical forms or better results in critical cases.

Pg 2 Line 26: breadth = width ??

---

## Author Comment (AC1) · 28 Oct 2018

article [utf8]inputenc

xcolor

**Response to Anonymous Referee 1**

hodges

October 2018

I thank the reviewer for the time and effort they put into reviewing this manuscript. The comments have helped me see the problems in the exposition of the original paper, and have inspired a substantial rewrite. The complete rewrite is uploaded as a supplement file with blue fonts used for major changed sections.

In the following response, I will first provide an overview of the changes that have been made in the manuscript, and then address the specific comments. Finally, I will provide a summary of where my response differs from the reviewer's comments and my reasons why.

Page and line numbers for the new manuscript are provided in brackets. Page and line numbers in parentheses are associated with the original manuscript.

**1 Overview of revision**

The rewrite has included a completely new Introduction section, which provides an overview of recent arguments about the SVE in large-scale hydrology. As a matter of personal preference in writing style and readability, the number of citations in the introduction is kept to a minimum and the reader is referred to succeeding sections for details.

The section that previously constituted the Introduction, which details the setting for understanding the SVE forms, is now a Motivation section. The changes in this section include some new citations on conservative vs. non-conservative forms [lines 25-30] and the presentation of the differential form of the equations using a free surface [eq 3], which is contrasted to the slope form of the equations [eq 2] and a new discussion of smoothness in the source term [lines 17-25]. There are also minor additions in this section to address specific comments (see below).

In the rewrite, the Background section has be subdivided into several topic areas - *Origination of the SVE, Preissmann v. Godunov methods,* and *"Well-balanced" problems and* $S_0$. Some minor additional text was added in a few places to clarify problems noted in the specific questions of the reviewer (see responses below). I disagree with the reviewer on what detail in model methods and discretizations should be presented in the Background, as discussed in the Disagreements section below.

For the Finite-Volume SVE section of the manuscript, the reviewer's comments indicated that I had not explained issues with the clarity that I had hoped. I have substantially revised the presentation in this section. The derivation is now divided into subsections to help delineate the arguments. The use of piezometric pressure and the fact that it is uniform along a vertical face is now discussed in a single section [pg 12,

lines 17 – 29] rather than implied in the equations. The explanation of the stair-step approximation, which resulted in the greatest confusion in the initial manuscript, has been improved with new figures and text from [pg 14, line 10] to [pg 19, line 16].

The section on Approximate Finite-Volume Forms in the manuscript has seen substantial additions, including 2 new figures [Figs 7 and 8] and text [pgs 21-22] to explain the meaning of the discrete quadrature of polynomial approximations of $\eta(x)$ and $\lambda(x)$.

The Discussion section of the manuscript now includes an enhanced discussion of $S_0$, the well-balanced condition, and smoothness [pg 29, lines 18 – 28] and [pg 30, lines 3 – 20].

**2 Specific questions and response**

Below are restatements of the specific questions of the reviewer and my responses.

REVIEWER: (Pg 2 Line 26): breadth = width ?

Yes. [pg 4, line 3] Breadth is used as synonym for width as using $B$ is less confusing than $W$ in the nomenclature. The latter can be confused with a vertical velocity. Some minor clarification in the text has been provided.

REVIEWER: Equation 8. Explain why f1 and f2 are necessary in eq 8 while it was not necessary in eq. 7.

**Done.** Now eq. 9. The $f_1$ and $f_2$ are intended as generic (unspecified) functions that have a similar form to Cunge-Liggett. This has been clarified by re-writing the equation

and revised text [pg 4 line 20 – pg 5, lines 1–4]

REVIEWER: Page 4 line 7. Review the frictional force per unit area

**Done.** Corrected to $F$ and clarified that its frictional force at the bottom of the channel. Thank you for your detailed reading that caught this typo [Pg 5, line 20].

REVIEWER: (pg 4 lines 17 – 29). Also review recent work using local inertia approximation from Bates et al. 2010 for large scale river networks.

**Partially Done.** [pg 6, lines 3 – 5] I have included the paper by Getirana et al (2017) as an example of the adaption of the local inertia approach to large scale modeling. I have also added a sentence referring to Hodges (2013) for arguments on SVE v. reduced physics models so as not to repeat the discussion here. I have not included a citation to the Bates et al (2010) paper as it was proposed as a 2D method for flooding rather than a 1D method for hydrology. Similarly, I have not included deeper references to the origins of the Muskingum or kinematic wave methods but instead have provided more recent references on their application. These few background sentences are to acknowledge that there are simplified forms for the SVE that are used in large-scale hydrology as the full SVE are considered to difficult to use. See also the section below on my disagreements with the reviewer.

REVIEWER: (pg 4 lines 17 – 29). Review work for short scale using HEC-RAS.

**Done.** [pg 6 lines 11 – 14]

REVIEWER: Please discuss explicit vs implicit schemes. Discuss local inertia explicit formulation. Review 6 point Abbot and Ionescu scheme (1967). Explain that Preissman scheme is a finite difference based on integral relations that improve conservation.

**Partially Done.** I have added an additional explanation [pg 6, lines 27 – 29] that clarifies why I am focusing only on the most fundamental issues (finite difference vs. finite volume) and not the details of the schemes (explicit vs. implicit). As the reviewer requested I have also added the Abbot and Ionescu paper, but only as one of the non-Preissmann finite-difference schemes. I have *not* added a discussion of 6 point v. 4 point finite-difference schemes. My reasoning is the Abbot and Ionescu is clearly related to the Preissmann scheme, but very different from Godunov methods, which is the key point for the background comparison. See also the section below on my disagreements with the reviewer.

REVIEWER: Please use more figures to help the reader to understand the paper

**Done.** See new Figures 1, 4, 5, 6, 7, and 8.

REVIEWER: Please use a figure to define the control volume

**Done.** See new Figure 1.

REVIEWER: (Page 8 line 2.) "This vector is local and change along the channel . . ..". Please use a figure to clarify it

**Done.** Refer to new figure 1 and revised text [pg 11, lines 2 – 3].

REVIEWER: (Page 8 line 25.) "It is known that . . ..". Please cite references showing these arguments.

**Done.** I have rewritten this section for clarity and provided reference to a typical work on flow in bends. [pg 11, lines 25 – 30]

[Figure]

REVIEWER: Figure defining the free surface slope n(x), these angles, etc.

**Done.** See new figure 1.

REVIEWER: Eq 25. Why P(x) is independent to z?

**Done.** Note that Eq. 25 is now Eq. 32 – this issue has been clarified with the definitional statement of the piezometric pressure as $\rho g \eta$, which is provided in the new subsection [pg 12, lines $17 - 30$]

REVIEWER: How the shape of cross section and its variability is considered?

This is the purpose of the stair-step model, the $A_R$ term, and $\lambda$. The section regarding the bottom pressure and topography have been significantly re-written to explain the $A_R$, as discussed in the Overview, above. See [pgs $14 - 19$]

REVIEWER: Page 12 equations 27, 28, 29. What is m? What lambda represents? Is it just a mathematical trick ? Or does it have any physical meaning? How it relates to zb(x) ? How it relates to river cross section shape and its variation along x? Why int (lambda)= 1 ? (eq 28)

**Done.** These are now the revised eq. $27 - 40$. Hopefully it is clear that $m$ is simply an index in a summation and the $\lambda$ is a weighting function that represents the distribution of the stair steps areas over the length of the bottom. The requirement that $\int \lambda \, dx = 1$ is a mathematical identity based on eqs $37 - 39$ so that the sum of all the stair-step areas is equal to the difference between $A_d$ and $A_u$.

REVIEWER: Equation 32. Ve is confusing here. It is volume but one can make a confusion with flow velocity. Please, use a good figure to define the control volume, what is Ve, Le, upstream and downstream cross section, etc.

[Figure]

**Done.** Figure 1 provides nomenclature. Note that this is now eq. 43. I agree that $V$ can be confusing, but I have been consistent throughout in using $V$ for volume in all the integration and $U$ or $u$ for velocity. As a matter of personal preference, I dislike use $\Omega$ for volume as is done by some authors.

REVIEWER: Equation 33. How gAn was obtained? Define how gAn is derived from P, using water surface n instead of depth H.

**Done.** Eq. 33 is now eq. 45. Additional text [pg 20, lines 15 – 19] has been added to explain the substitutions from eq. 44 to 45. The relationship between piezometric pressure and the free surface is given in the new subsection [pg 12, lines 16 – 30] and eq. 23.

REVIEWER: Section 4. Please define the meaning of lambda function. It is important to understand the choice for polynomial approximations.

**Done.** See eq. 38 and discussion on [pg 19, lines 8 – 17]

REVIEWER: Please show in a figure the approximations T(0,0), T(1,0), T(2,0), T(1,1) of n(x) and lambda(x) or zb(x).

**Partially Done.** I disagree with the reviewer on the usefulness of a figure illustrating the difference between zeroth, first, and second-order polynomial interpolation. However, the reviewer's comment has pointed out that I was not clear on explaining what was being done. I have extensively re-written the description of the approximation approach, beginning [pg 21, line 22] and ending on [pg 24, line 5]. I have included two new figures that I think are helpful in illustrating key concepts. Also, new text has been added on [pg 26, lines 4 – 6] and [pg 27, lines 1 – 3] to clarify the meaning of the $T_{e(1,1)}$ approximation.

REVIEWER: Equation 60. Check it. I guess the last term is g.Ae.Le.Sfe.

**Done.** This is now eq 72. Thank you for catching the typo.

REVIEWER: Pg 19, line 22. "S0 brings a host of problems". What problems? Please show it based on past publications.

**Done.** See revised explanation [pg 29, lines 22 – 28]

REVIEWER: Eq. 62. Not clear

**Done.** This was speculation on future uses. As it was not clear I have removed the equation and revised the discussion [pg 30, lines 21 – 28].

REVIEWER: Final discussions and conclusions. In my view, the paper would be much more convincing if a few examples showing that application of the new forms of SV equations provide the same results as other classical forms or better results in critical cases

**Not done.** I disagree with the reviewer on this point. Please see the discussion below.

**3 Disagreements and rebuttal**

The reviewer has made suggestions on three major points where we disagree: (1) extended background on reduced-physics models, (2) extended background on numerical discretization schemes, and (3) examples of numerical implementation of

the new governing equation form and comparison against other methods. Although I appreciate the reviewer's viewpoint and the usefulness of each of these ideas, adequately addressing each would make this paper overly-long and substantially diffuse the principle message of the paper. Each of these issues is addressed separately below.

Extended background on reduced-physics models

My interpretation of the reviewer's comments is that they believe there should be greater comparisons/background with reduced-physics methods and/or some attempt to demonstrate that there are problems with the SVE in large-scale models. That is, the reviewer appears to want me to prove that there is a problem with reduced-physics models before I can present a potential solution. Unfortunately, our present approach to scientific publication makes it difficult (if not impossible) to present a paper that shows the difficulties of using SVE for large-scale systems – our journals are biased towards presentation of success rather than analyses of failure. I have added citations to forum papers where this issue has been discussed, but I believe the overall issue is beyond the scope of this manuscript and is not necessary to put the present work into context. That is, the theoretical derivation of the new finite-volume method is a new advance whether or not reduced-physics models are adequate. Thus, an extended discussion of reduced-physics models will detract from the focus of the paper on showing a different form of governing equations that are compared to other full SVE governing equations.

As a matter of philosophy, in my view the SVE are the fundamental governing equations for river flow so the fact that they are *not* used for large-scale hydrological modeling is all the evidence that we need that there are problems with the SVE. It should be incumbent

on those presenting reduced-physics models to show that their neglect of dynamics is well founded rather than merely convenient. Unfortunately, reduced-physics models typically rely on comparison of calibrated results to observations rather than analyses showing that the missing physics are unimportant. Thus, results from reduced-physics models cannot distinguish whether they correctly neglected unimportant physics or merely calibrated an important (but neglected) piece of physics within another term.

Extended background on numerical discretization schemes

I do not think that a detailed discussion of explicit vs. implicit methods, the number of stencil points, or the motivation for the Preissmann stencil relative to other stencils etc. is appropriate for this paper. I am presenting a new form of the governing equation for momentum – i.e. an integral form that is suitable for finite-volume discretization. Thus, the Motivation and Background sections focus on the prior differential and integral forms of the SVE rather than on the details of the numerical schemes used to solve the equations. I have only discussed different numerical techniques (Preissmann and Godunov) to provide a convenient way to classify the basic methods and introduce how others have dealt with problems of conservation, well-balanced methods, and bottom slope. All of these issues have been the subjects of discussion within the cited literature.

Examples of numerical implementation of the new governing equation form and comparison against other methods

I agree that the the manuscript would be "more convincing" of the value of the new equation form with one or more discrete numerical examples. However, it would also be twice as long. This would be a problem as the paper has already reached 12000 words. To keep the paper within reasonable length, I am focused only on the

general formulation of the equations rather than the detailed numerical implementation.

This manuscript actually evolved from a working paper on a new numerical scheme with a particular discretization method; however, whereas most numerical methods papers can introduce the differential or integral form of the governing equations with a single paragraph, it became clear that the approach I wanted to take would require many pages to present and justify the new integral finite-volume form. As the manuscript grew in length, it soon became clear that there were really two papers: (1) the theoretical derivation of the new integral form, presented here, and (2) a numerical implementation of the new form and analyses of its behavior – which is under review at another journal.

My argument for *not* presenting an example numerical scheme and results is that *whether the new form is valid or not is a matter of mathematical derivation and proof, which does not depend on a specific implementation.* Similarly, de Saint-Venant's paper was a valid and useful contribution despite the inability to actually solve the equations in 1871.

**4 A final note**

Again, I thank the reviewer for their thoughtful comments. Their opinions and ideas have forced me to revisit my methods of presentation and provide clearer explanations for the mathematics of the derivation. I hope they find the end result pleasing and recognize that they have made a positive addition to my work.

**Supplement:**

[revised manuscript text omitted]

The new conservative form of the SVE is developed with a goal of addressing challenges associated with modeling large-scale 1D flow network systems. In the process of developing the new form, we will encounter a philosophical question as to whether the primary vertical variable in a large-scale network solution should be the depth ($H$) or the water surface elevation ($\eta$). Despite our prior work with $H$ primacy (Liu and Hodges, 2014), we shall see that there are advantages to using $\eta$ as it is identical to the piezometric pressure, which is uniform over a channel cross-sectional area. The new derivation herein provides interesting possibilities for analytically including hyperresolution bathymetric knowledge while retaining larger computational elements for large-scale modeling. The interaction of subgrid-scale topography with subgrid free-surface gradients is handled in a new integrated piezometric pressure term that arises in the derivation. 
[revised manuscript text omitted]
 = 0$, where the free surface should be exactly flat and $S_f = 0$. Achieving the simple result of $Q = 0$ for a flat free surface with the Cunge-Liggett form requires

$$\frac{\partial I_1}{\partial x} = I_2 + AS_0 \iff H + z_b = \text{constant} \tag{12}$$

which implies the Cunge-Ligget form is only well-balanced if the geometry meets the following identity at every possible water surface level:

$$\frac{\partial}{\partial x} \int_{z_b(x)}^{z_R} (z_R - z) B \, dz - \int_{z_b(x)}^{z_R} (z_R - z) \frac{\partial B}{\partial x} \, dz = AS_0 \quad : \quad z_b < z_R \leq \eta_{max} \tag{13}$$

where $\eta_{max}$ is the maximum water surface elevation, $z_b(x)$ is the local channel bottom elevation, and $z_R$ encompasses all possible water surface elevations. Clearly, designing a numerical scheme that *exactly* preserves this relationship for non-uniform

channels is a challenge, as evidenced by the breadth and complexities of studies focused on this issue (e.g., Audusse et al., 2004; Bollermann et al., 2013; Bouchut and Morales de Luna, 2010; Castro Diaz et al., 2007; Crnković et al., 2009; Kesserwani et al., 2010; Kurganov and Petrova, 2007; Li et al., 2017; Liang and Marche, 2009; Perthame and Simeoni, 2001; Xing, 2014). Failure to satisfy the well-balanced criteria results in models that generate spurious velocities; i.e., a mismatch in eq. (13) indicates that the numerics provide momentum sources/sinks that are functions of channel shape and discretization rather than flow physics. An interesting approach to this problem was developed by Schippa and Pavan (2008) where eq. (12) is used to replace $I_2 + AS_0$ in the source term with $\partial I_1 / \partial x$ evaluated for a horizontal surface. Their approach ensures that *any* discretization will be well-balanced for a zero-velocity flow.

The work of Schippa and Pavan (2008) and a review of other works on well-balanced schemes provides us a key insight: the principal challenge for obtaining a well-balanced method is the channel bottom slope, $S_0$, which is often sharply varying or even discontinuous in a natural system. Furthermore, as a geometrical property, $S_0$ should be independent of the cross-sectional flow area ($A$), and yet is forced to be discretely related through eq. (12). If we take this idea a step further, we can argue that the fundamental problem with the Cunge-Liggett form is that the physical forces that alter momentum (gravitational potential and hydrostatic pressure) are arbitrarily separated so that one is wholly within the source term and the other has an *ad hoc* split between conservative flux and source terms. Thus, we return to the idea put forward in the Introduction that we should consider the free surface elevation (piezometric pressure) instead of the water depth (hydrostatic pressure) as our primary forcing gradient.

In the next section, we shall see how the ideas of shifting portions of the total piezometric pressure from source to flux can be used to develop a rigorous finite-volume form of the SVE that is simpler those based on the Cunge-Liggett form.

**4   Finite-volume SVE with minimal approximations**

**Continuity**

Although we are focused on the momentum equation, for completeness we will start with continuity. The general arrangement of the control volume for an irregular channel and the vectors used in the following discussion are illustrated in Fig. 1. Applying only the incompressibility approximation for a uniform density fluid, the volume-integrated continuity equation is

$$\frac{\partial V}{\partial t} + \oint_S u_k n_k \, dA = S_V \tag{14}$$

where the Einstein summation convention is applied on repeated subscripts, $u_k$ is a vector velocity, and $n_k$ is a unit normal vector defined as positive pointing outward from a control volume $V$, and $S_V$ is a volume source ($S_V < 0$ for a sink).

A semi-discrete finite volume representation of continuity can be directly written as

$$\frac{\partial V_e}{\partial t} = Q_u - Q_d + q_e L_e \tag{15}$$

where $Q$, and $L$ represent the flow rate and element length, and subscripts $e, u,$ and $d$ denote characteristic values for the control volume element, nominal upstream face, and nominal downstream face, respectively. Here we use the "nominal" flow direction

[Figure]

**Figure 1.** General arrangement of control volume element ($V_e$) and its neighbors for irregular channel. Unit normal vectors $n_k$ are always perpendicular to cross-sectional areas ($A_u$, $A_d$) and pointing outward from control volume. The element length ($L_e$) is measured along the channel. Unit vectors $\hat{i}_k$ are coincident with the free surface slope in the streamwise direction and can be defined as local continuous functions. The velocity vector $u_k$ is approximated as parallel to $\hat{i}_k$. The angle measured from $n_k$ to $\hat{i}_k$ is $\psi$. The free surface elevations $\eta_u$, $\eta_e$, and $\eta_d$, are cross-section uniform elevations at the upstream face, for the element center, and the downstream face, respectively. Note that volume is $V$ throughout the derivations with $u$ and $U$ used for continuous and spatially-averaged velocities, respectively.

as the global downhill direction of the channel that is assigned at the network level. The average lateral inflow per unit length is $q_e$, and a flow $Q > 0$ is from the upstream to downstream direction. Reversals of flow from the nominal flow direction are handled with $Q < 0$.

**Momentum**

The control-volume form of the Navier-Stokes momentum equation in a direction defined by unit vector $\hat{i}$ in a Cartesian frame is

$$\frac{\partial}{\partial t}\int\limits_V u_{\hat{i}}dV + \oint\limits_S u_{\hat{i}}u_k n_k\,dA + \frac{1}{\rho}\oint\limits_S p\hat{i}_k n_k\,dA = \oint\limits_S \nu\frac{\partial u_j}{\partial x_k}\hat{i}_j n_k\,dA + \int\limits_V g_{\hat{i}}dV \tag{16}$$

5 where $u_{\hat{i}}$ is a velocity vector component in the $\hat{i}$ direction, $u_k$ are velocity components along the Cartesian axes, the gravity vector is $g_{\hat{i}}$ in the $\hat{i}$ direction, $\nu$ is the kinematic viscosity, and $p$ represents the thermodynamic pressure. Note that this formulation can be related to any arbitrary Cartesian axes. In many common derivations, $\hat{i}$ is approximated as coincident with an $x$ axis that is a *horizontal* vector in the streamwise direction. In the following, we will show that this approximation is not required. Instead, we treat this as a simplification that can be applied to the final equation form.

10 ## Advection terms

The $\hat{i}$ direction for momentum, eq. (16), is a vector associated with the $u_{\hat{i}}$ velocity component, which is not necessarily coincident with the normal vector $n_k$ at a surface of a finite volume (in contrast to the case where $\hat{i}$ is taken as horizontal). For a gradually-varying open-channel flow we can take the $\hat{i}$ vector as the nominal downstream direction along the channel centerline described by a vector *that lies along the free surface*, as illustrated in Fig. 1. Thus, this vector is local (as opposed
15 to being forced into coincidence with a Cartesian axis) and changes along the channel with the slope of the free surface. It follows that the discrete control volume formulation is globally exact as $V_e \to 0$. In contrast, a derivation that takes $\hat{i}$ as a vector in the horizontal direction has a momentum conservation error proportional to $\cos\psi$, where $\psi$ is an angle between the horizontal vector and the free surface; such an error does not vanish as $V_e \to 0$ unless the free surface is flat across the length of the element. This idea helps illustrate one of the subtle advantages of the Godunov approach in which the channel is imagined
20 as having a piecewise flat free surface: $\cos\psi = 1$ is then an identity within the approximation of the physics, rather than as an approximation within the mathematics.

The upstream element face is required to be vertical and normal to the smooth channel centerline in a horizontal plane, as illustrated in Fig. 1. The free surface at the centerline has an angle of $\psi(x)$ to the horizontal so that the discrete nonlinear momentum term in the $\hat{i}$ direction across the upstream face is formally

$$\int\limits_{A_u} u_{\hat{i}}u_k n_k dA = \beta Q_u U_u \cos\psi_u \tag{17}$$

where $u$ subscripts indicate the upstream face (rather than vector components), $\hat{i}$ is defined normal to the surface, $U$ is the average velocity normal to the face (which implies $Q = UA$), and $\beta$ is the momentum coefficient for the streamwise velocity, defined as

$$\beta \equiv \frac{1}{A\left[U(x)\right]^2}\int\limits_A \left[u(x,y,z)\right]^2 dA \tag{18}$$

Note that the convective term of eq. (17) does not contain any approximations.

It is convenient to let the $M_e$ represent the integrated sources (+) and sinks (−) of momentum per unit mass in the finite-volume element associated with the $q_e L_e$ lateral fluxes of eq. (15). Furthermore, *if the channel is straight* so that $\hat{i}$ at the upstream face is parallel to $\hat{i}$ at the downstream face, it follows that an exact finite-volume integration is

$$\quad \oint_S u_{\hat{i}} u_k n_k \, dA = -\text{sgn}(U_u)\,\beta_u\, Q_u\, U_u \cos\psi_u + \text{sgn}(U_d)\,\beta_d\, Q_d\, U_d \cos\psi_d - M_e \tag{19}$$

where $\text{sgn}(U)$ is a sign function returning +1 for $U > 1$, −1 for $U < 1$, and 0 for $U = 0$. For the more general case where the channel is curved between the upstream and downstream faces (as in Fig. 1), the above integration becomes an approximation – i.e. the effects of cross-channel gradients on the nonlinear advective term are neglected in the discrete form. For present purposes, the use of a $\hat{i}$ direction that follows the curving channel implies that pressure is perfectly redirecting momentum

[revised manuscript text omitted]

$$\int\limits_{A_d} \tilde{P}\hat{i}_k n_k \, dA = +P_d A_d \cos\psi_d + \cos\psi_d \int\limits_{A_d} \breve{P}(x,y,z) \, dA \qquad (27)$$

Note that using the piezometric pressure instead of the hydrostatic pressure ensures that the only term requiring a formal integration at the upstream and downstream surfaces is the non-hydrostatic pressure.

**5 Pressure on bottom**

The third pressure term in eq. (21) is more challenging than the pressure on the flow faces. To arrive at a simpler formulation, we note the channel bottom has some local angle $\hat{\theta}(x,y)$ to the horizontal (i.e., the local bottom slope angle), which is measured clockwise from a downstream-pointing horizontal line to the bottom as illustrated in Fig. 2. Here we need to assume that cross-channel variations can be represented with a characteristic slope, $\theta(x)$, which adequately represents the true variability of $\hat{\theta}(x,y)$ across the channel at location $x$. The modified pressure ($\tilde{P}$) at the bottom is, by definition, normal to the bottom, so where $\theta(x) = \psi(x)$ the bottom is exactly parallel with the free surface and the pressure at the bottom is exactly normal to $\hat{i}$ and thus normal to the streamwise flow. It follows that under this restricted condition the integral of $P\hat{i}_k n_k$ over $A_B$ is identically zero and the third pressure term in eq. (21) vanishes – that is, gradients in pressure that are normal to the streamwise direction cannot alter momentum. However, for the more general case when the bottom and the free surface are not precisely parallel, $\tilde{P}$ at the bottom can be decomposed into a component acting normal to the free surface and a component parallel to the free surface (i.e., a modified bottom pressure contribution acting along the $\hat{i}$ flow direction), as shown in Fig. 2. The effect of the bottom topography acting on the fluid can be represented as

$$\int\limits_{A_B} \tilde{P}\hat{i}_k n_k \, dA \approx -\int\limits_{A_B} \left|\tilde{P}(x,y,z)\right| \sin\left[\theta(x) - \psi(x)\right] dA \qquad (28)$$

where the sign of the term is selected for consistency with the $\hat{i}$ nominal downstream direction and the definitions of $\theta$ and $\psi$ to provide for the correct sense in either diverging ($\theta > \psi$) or converging ($\theta < \psi$) conditions. The condition $\theta > \psi$ implies that the fluid pressure at the bottom (normal to $\theta$) will have a component in the upstream direction when decomposed into vectors normal and tangent to the $\psi$ of the free surface, which means the force of the bottom on the fluid is acting downstream, with a normal vector ($n_k$) that points outward from the element (e.g., upstream). Note that this approach allows consideration of adverse free surface gradients ($\psi < 0$) and adverse slope angles ($\theta < 0$) without any special treatment.

To develop a simpler implementation of the bottom pressure term, we need an approximation for the contribution of the vector component of pressure along the bottom that is aligned with the $\hat{i}$ direction of the free surface. That is, we can neglect the bottom pressure contribution that is perpendicular to $\hat{i}$ since it cannot contribute to streamwise momentum. We introduce a conceptual model with the channel bottom imagined as $m = 1...N$ stairsteps where the treads are locally parallel to the free surface and the risers are normal to the free surface, as illustrated in Figs. 3 through 5.

Clearly, as $N \to \infty$ we will recover a continuous approximation so there is no need to actually consider the discrete stair steps in a solution method – the stair-steps are merely to illustrate what is otherwise a mathematical abstraction in vector

[Figure]

**Figure 2.** Pressure decomposition to obtain streamwise contribution.

[Figure]

**Figure 3.** Stair-step approximation for pressure along bottom where tread is parallel to the free surface and riser is perpendicular for linear slopes of both bottom and free surface.

calculus. We can imagine the stair-step risers as thin planar strips across the entire wetted perimeter that provide a discrete representation of irregular channel cross-section structure, as illustrated in Fig. 6. The only approximations needed for this conceptual model are: (1) the free surface at longitudinal position $x$ is uniform over the cross-section and aligned with $\hat{i}(x)$, and (2) the channel topography at longitudinal position $x$ has a characteristic slope angle $\theta(x)$ that is uniform over the cross section. Since the stair-step treads are (by definition), normal to the modified pressure above, it follows that the only pressure contributions to the momentum in the streamwise $\hat{i}$ direction are on the risers, with individual areas $A_{R(m)}$ for $m = 1...N$ stairsteps. Because the pressure contribution for increasing cross-sectional area (i.e., steps down as in Fig. 3) will be opposite of the pressure contribution for decreasing cross-sectional are (i.e., steps upward, see downstream section of Fig. 5), it is convenient to introduce a function $\gamma_{(m)} = \pm 1$ to account for the change of sign needed for the direction of the pressure force.

[revised manuscript text omitted]

$$A_d\cos\psi_d - A_u\cos\psi_u \approx \sum_{m=1}^{N} A_{R(m)} \tag{35}$$

However, in the limit as $N \to \infty$ we have a single value of $\cos\psi$ at any point along a smooth free surface so that the continuous form provides an identity:

$$A_d\cos\psi_d - A_u\cos\psi_u = \int_L A_R(x) \, dx \tag{36}$$

To generalize the above for $A_d < A_u$, we can use the $\gamma(x) = \pm 1$ that was introduced for the pressure direction in eq. (30). Values of $\gamma(x) = +1$ indicates the cross-section area is increasing across location $x$ in the streamwise direction (as in Figs. 3 and 4), whereas $\gamma(x) = -1$ indicates the cross-sectional area is decreasing (as in the latter portion of Fig. 5). It follows that

$$A_d \cos\psi_d - A_u \cos\psi_u = \int_L \gamma(x) A_R(x) \, dx \tag{37}$$

5   is an identity that should be satisfied for a control volume with any continuous, smooth bottom topography and free surface.

To handle $A_R(x)$ in the pressure term of eq. (32), we introduce a quadrature function $\lambda(x)$, defined as

$$\lambda(x) \equiv \frac{\gamma(x) A_R(x)}{A_d \cos\psi_d - A_u \cos\psi_u} \tag{38}$$

Note that eq. (37) implies the identity:

$$\int_L \lambda(x) \, dx = 1 \tag{39}$$

10   Using the above in the first term on the RHS of eq. (32), we obtain

$$\int_L \gamma(x) \, P(x) \, A_R(x) \, dx = (A_d \cos\psi_d - A_u \cos\psi_u) \int_L P(x) \, \lambda(x) \, dx \tag{40}$$

Thus, the introduction of $\lambda$ allows us to extract a multiplier from the control volume integral of the bottom pressure. As a result, $\lambda(x)$ is merely a distribution, or "weighting" function for integration of $P(x)$. The full bottom pressure term, eq. (32), can be written as

15   $$\int_{A_B} \tilde{P} \hat{i}_k n_k \, dA = -(A_d \cos\psi_d - A_u \cos\psi_u) \int_L P(x)\lambda(x) \, dx - \int_L \gamma(x) \int_{A_R} \check{P}(x,y,z) \, dA \, dx \tag{41}$$

Note that $\lambda$ weighting cannot be readily applied the non-hydrostatic term because the non-hydrostatic pressure on the bottom has spatial distributions in both the vertical and across a channel that cannot be assumed negligible; hence we cannot pass $\check{P}$ through the $A_R$ integration as was done in eq. (31) for $P$.

We can think of $\lambda(x)$ as a weighting function of the conceptual stair-step riser areas over the control-volume length, which
20   controls where the piezometric pressure gradients have their greatest effect. For example, in Fig. 3 the stair-step risers are uniformly distributed such that we can use $\lambda(x) = L^{-1}$, which meets the identity requirement of eq. (39). In contrast, Fig. 4 implies $\lambda(x)$ is perhaps a quadratic function. Figure 5 presents a challenge as $\lambda(x)$ should reverse in sign between the upstream and downstream faces. A key point in this new finite-volume derivation is the selection of $\lambda$ functions provides a more general discrete control over the representation of the free surface and bottom topography within a control volume. This can
25   be contrasted to the Godunov approach that approximates a control volume as piecewise uniform for both the bottom elevation and free surface elevation over a control volume (see Section 3). Several discrete approaches to approximation of $\lambda(x)$ will be examined in Section 5, although the full consequences and utility of the $\lambda$ approach will require more extensive investigation for both theoretical limitations and practical discretization schemes.

**Combining pressure terms**

In summary, the pressure terms of eq. (21) can be written using eqs. (26), (27) and (41), resulting in:

$$\frac{1}{\rho}\oint_S \tilde{P}\hat{i}_k n_k\,dA = -\frac{1}{\rho}A_u P_u\cos\psi_u + \frac{1}{\rho}A_d P_d\cos\psi_d - \frac{1}{\rho}\left(A_d\cos\psi_d - A_u\cos\psi_u\right)\int_L P(x)\lambda(x)\,dx$$

$$-\frac{\cos\psi_u}{\rho}\int_{A_u}\check{P}(x,y,z)\,dA + \frac{\cos\psi_d}{\rho}\int_{A_d}\check{P}(x,y,z)\,dA - \frac{1}{\rho}\int_L\gamma(x)\int_{A_R}\check{P}(x,y,z)\,dA\,dx \tag{42}$$

where the last three terms are the non-hydrostatic pressure effects that are typically neglected in the SVE.

**Viscous term**

The remaining term in eq. (16) is the viscous term, which is treated as an empirical function in all but the most highly-resolved models of simple systems (note that Decoene et al. (2009) provides a comprehensive and rigorous approach for friction that has not yet been fully considered in SVE models). For the present purposes, we will retain the simple friction slope form with an assumption of linear behavior over space, i.e.

$$\oint_S \nu\frac{\partial u_j}{\partial x_k}\hat{i}_j n_k\,dA = -g\int_{V_e} S_f(x)\,dV \approx -gV_e S_{f(e)} \tag{43}$$

where $S_{f(e)}$ is the average friction slope over the control volume $V_e$.

**Finite-volume for momentum**

Putting together the above, eq. (16) can be written in a finite-volume form as

$$\frac{\partial}{\partial t}(U_e V_e) = \mathrm{sgn}(U_u)\beta_u Q_u U_u\cos\psi_u - \mathrm{sgn}(U_d)\beta_d Q_d U_d\cos\psi_d + \frac{1}{\rho}A_u P_u\cos\psi_u - \frac{1}{\rho}A_d P_d\cos\psi_d$$

$$+\frac{1}{\rho}\left(A_d\cos\psi_d - A_u\cos\psi_u\right)\int_L P(x)\lambda(x)\,dx - \frac{\cos\psi_u}{\rho}\int_{A_u}\check{P}(x,y,z)\,dA + \frac{\cos\psi_d}{\rho}\int_{A_d}\check{P}(x,y,z)\,dA$$

$$-\frac{1}{\rho}\int_L\gamma(x)\int_{A_R}\check{P}(x,y,z)\,dA\,dx - gV_e S_{f(e)} + M_e \tag{44}$$

where $U_e$ is the element velocity and $V_e$ is the element volume. Note that $Q > 0$ and $U > 0$ imply flow in the nominal downstream direction, whereas $Q < 0$ and $U < 0$ imply flow in the nominal upstream direction. At this point we have introduced only five approximations: (1) uniform-density incompressibility, (2) the effect of channel curvature is either negligible or handled in an empirical viscous term, (3) the cross-channel variability in the free-surface slope is negligible, (4) the cross-channel variability in the bottom slope $\hat{\theta}(x,y)$ at any cross-section can be characterized by a single slope angle $\theta(x)$, 
[revised manuscript text omitted]

$$-gA_u\eta_u\cos\psi_u + gA_d\eta_d\cos\psi_d - \frac{\cos\psi_u}{\rho}A_u\breve{P}_u + \frac{\cos\psi_d}{\rho}A_d\breve{P}_d$$

$$= g\left(A_d\cos\psi_d - A_u\cos\psi_u\right)\int_L \eta(x)\lambda(x)\,dx + \frac{1}{\rho}\int_L \gamma(x)\int_{A_R}\breve{P}(x,y,z)\,dA\,dx - gV_e S_{f(e)} + M_e \tag{71}$$

where $\breve{P}_u$ and $\breve{P}_d$ are the average non-hydrostatic pressures on the upstream and downstream cross-sectional areas. In this finite-volume form, the only approximations introduced are: (1) uniform-density incompressibility, (2) the effects of momentum redirection around bends is either negligible or is handled in friction terms, (3) the cross-channel variability in the free-surface slope is negligible, (4) the cross-channel variability in the bottom slope at any cross-section can be reasonably represented by a single slope angle $\theta(x)$, 
[revised manuscript text omitted]

---

## Referee Comment (RC2) · Anonymous Referee #2 · 9 Nov 2018

The paper deals with the derivation of new forms of the momentum equation for onedimensional open-channel flow, suitable for the implementation in hydrologic and urban drainage finite volume models. I enjoyed reading the paper

I have read the paper first (with pleasure), then the comments by referee #1, finally the response of the Author. I was satisfied with the response of the Authors but, unfortunately, I was unable to see the supplement with the updated version of the paper. My review thus relies on the original version of the manuscript.

As my general impression was in fair agreement with that of referee #1, and given the positive response by the Authors, I am reporting here only some additional comments

regarding some unclear aspects (that maybe have already been addressed in the updated version of the manuscript). Hence, I am looking for seeing the revised version of the manuscript.

**Major comments**

-p.8, eq. 17: I have some doubts on this equation. For what I understand, the LHS term of Eq. 17, in which appears the vector  $u_{\hat{i}}$ , is in vector form. The RHS terms seem to be scalars (i.e., projections). Unfortunately, the versor  $\hat{i}$  is allowed to change between the upstream and downstream sections of the finite volume (only his slope is allowed to change, of course, as his horizontal direction is assumed constant here). In the RHS, the first term is projected along  $\hat{i}_u$ , the second term is projected along  $\hat{i}_d$ , the third term I do not know. Please clarify.

-p.9, I.21-ff: This reasoning seems to be tailored for rectangular cross-sections with the breadth B=constant. It is not so intuitive (to me) to extend it to cross-sections of general shape. If the breadth changes along the control volume, the side pressure has a component along the channel axis too. How it this considered? At p.11, I.1 I see "bottom pressure term", but the wet boundary of the channel is not only its bottom.

-p.12, l.10-ff: This reasoning should apply also to the cross-sectional area, not only to the bottom elevation (if the cross section is not prismatic, see the comment above).

-p13, I. 4: the implications of the second approximation deserve some additional comments. Does this approximation mean that the derived equations are only suitable for (smooth) subcritical flows? The power of FV scheme is the ability to handle rapidly varying flows, discontinuities and shock waves.

-p.19, I.28: When the cross-sections are broadly spaced... This is not a trivial issue. In fact, in hydrology and urban drainage applications is generally difficult to include a great number of (close to each other) cross sections. However, the use of broadly spaced cross sections conflicts with the 2nd geometric restriction (bottom elevation
varying monotonically within the control volume). Moreover, how to find a tradeoff between broadly spaced cross-sections and an effective representation of convective accelerations? (this last point is maybe less important when considering smooth flows).

Minor comments

- -p.2, I.1: form of momentum equation
- -p.2, I.27: opposite to?
- -p.4, I.26: parenthetical citation "(Burger et al., 2014)".
- -p.7, I.6: simpler than those
- -p.8, I.9: vertical and normal to the horizontal projection of the mean flow

-p14, l.11: an, not and.

-p.18, I6-ff: already said at the beginning of p.13. (In general, the discussion seems more a summary than a discussion).

-p.19, I.25: delete the repeated "as".

HESSD
**Discussion** paper

---

## Author Comment (AC2) · 18 Dec 2018

xcolor

From the author: My thanks to Reviewer 2 for the constructive comments. I believe that responding to these comments has significantly improved the paper.

I have added a supplemental file that is a markup using a blue font for all the major changes.

REVIEWER: Comment 1: -p.8, eq. 17: I have some doubts on this equation. For what I understand, the LHS term of Eq. 17, in which appears the vector $u_i$, is in vector form. The RHS terms seem to be scalars (i.e., projections). Unfortunately, the vector $\hat{i}$ is allowed to change between the upstream and downstream sections of the finite volume (only his slope is allowed to change, of course, as his horizontal direction is assumed constant here). In the RHS, the first term is projected along $\hat{i}_u$, the second term is projected along $\hat{i}_d$, the third term I do not know. Please clarify.

Response: This is now eq. 19. You've pointed out an area of the paper that really wasn't very clear. The advective terms are not the prime focus of the paper, so I wasn't quite as detailed as I should have been. I have revised the discussion substantially and added Appendix A to clarify the details. The complexity arises because of the subtle distinctions between the streamwise velocity $(u_{\hat{i}})$ that is at an angle to the control volume face and the normal velocity $u_k n_k$ that the provides the flux $Q$. The key point is that for a simple straight segment with a linear free surface, the control volume formulation integrates to exactly eq. 19, and for a curving system and/or non-linear free surface the equation is approximate, but exact as $L \to 0$. The discussion has been substantially revised and the definition of $M_e$, the source/sink terms, moved from above the equation to below the equation, which makes it easier to find.

REVIEWER: Comment 2: p 9, l.21-ff: This reasoning seems to be tailored for rectangular cross-sections with the breadth B=constant. It is not so intuitive (to me) to extend it to cross-sections of general shape. If the breadth changes along the control volume, the side pressure has a component along the channel axis too. How it this considered? At p.11, l.1 I see "bottom pressure term", but the wet boundary of the channel is not only its bottom.

Response: I did not intend "channel bottom" to apply only to prismatic shapes where

the flat bottom is distinct from sides. I find that conceptual models requiring a separation of sides and bottom to be troubling as the transitions in a natural channel seem to be arbitrary. The intent (although I clearly missed the mark in the explanation) is to treat general bottom topography with some local surface normal $\hat{n}$, from which only the components in the streamwise direction can have impact on the flow. This applies even when the bottom is a side. In effect, when the channel side changes from uniform to increasing breadth the surface normal changes from pointing across the channel to pointing slightly downstream. This is translated into a stair-step area that provides a pressure contribution in the streamwise direction. I have substantially revised the discussion under the section **Pressure on bottom** – including renaming the section **Pressure on bottom topography** to emphasize the point. I think the addition of Fig. 4 and the revised discussion helps illustrate how widening of the channel is represented as part of the 3D stairstep structure. I have introduced a further way of envisioning the bathymetry as rectilinear bricks that may help seeing how the concept extends in 3D.

REVIEWER: Comment 3: p.12, l.10-ff: This reasoning should apply also to the cross-sectional area, not only to the bottom elevation (if the cross section is not prismatic, see the comment above).

Response: This discussion, starting before Eq. 28 has been completely rewritten to be clear that it applies to general topography and not simply to prismatic sections.

REVIEWER: Comment 4: p13, l. 4: the implications of the second approximation deserve some additional comments. Does this approximation mean that the derived equations are only suitable for (smooth) subcritical flows? The power of FV scheme is the ability to handle rapidly varying flows, discontinuities and shock waves.

Response: The second approximation you're referring to is that "effects of momentum redirection around bends are handled in friction and non-hydrostatic pressure terms". As a direct answer to your question: this approximation (although poorly described) does not have any effect on whether the derived equations can be used for only smooth subcritical flows – the question of suitability for transcritical flows depends on the discretization of the scheme rather than the governing equations. The point I was trying to make (albeit badly,) is that this is a fundamental approximation in all Saint-Venant solutions of real river channels that have curvature – but is ignored in almost all the literature. Momentum in the $x$ direction doesn't just become momentum in the $y$ direction around a bend without the intervention of a pressure gradient. And yet, when we solve equations that "unwrap" the river with a streamwise coordinate system, we are assuming that that pressure-induced redirection is immaculate and without loss of momentum.

To try to clarify, the statement in question has been re-written as "the effects of momentum redirection around bends is either negligible or is handled in friction terms" so as to avoid prolonged discussion of the role of cross-channel gradients of hydrostatic pressure and non-hydrostatic pressure in momentum re-direction. In the section on the advection terms (where the approximation is originally introduced), the following is the revised discussion: "the use of a gradually-varying streamwise $\hat{i}$ direction implies that pressure is perfectly redirecting momentum through bends and aligning the momentum with the free surface. These are (generally) unstated approximations used in common 1D SVE formulations. However, it should be noted that this perfect momentum redirection is not precisely correct; e.g., secondary circulation in bends affects bed shear, velocity distribution, and frictional losses (Blanckaert and Graf, 2004)."

REVIEWER: Comment 5: -p.19, l.28: When the cross-sections are broadly spaced... This is not a trivial issue. In fact, in hydrology and urban drainage applications is

generally difficult to include a great number of (close to each other) cross sections. However, the use of broadly spaced cross sections conflicts with the 2nd geometric restriction (bottom elevation varying monotonically within the control volume). Moreover, how to find a tradeoff between broadly spaced cross-sections and an effective representation of convective accelerations? (this last point is maybe less important when considering smooth flows).

Response: I entirely agree. Broadly-spaced cross-sections are not a trivial issue and there are many trade-offs to be considered. However, to delve into this issue in the Discussion section of the paper would probably be too speculative given that what is practical depends quite strongly on the discretization of the model rather than specifically the governing equations - with the proviso that $S_0$ is a major issue of the governing equations. From my perspective, if a modeler has the cross sections that shows the river is non-monotonic then those cross sections should be used in the model to avoid non-monotonic finite volumes. My personal opinion is that the difficulty in including a large number of cross sections in hydrological and urban drainage models is due to (1) poor model construction that makes inefficient use of parallel computing power, and (2) the introduction of non-smooth $S_0$ that makes it difficult to converge a solution where cross sections are close together. There are certainly some data problems associated with getting enough cross sections, but I've seen too many instances of cross sections that are close together being thrown out simply because the model couldn't converge. The idea that we throw out good data because it doesn't work in our models is really quite troubling. I've added some further discussion on the use of $S_0$ and why we shouldn't use it – I hope you find this useful.

REVIEWER: Minor comments (not repeated here)

Response; All the minor comments have been fixed, with the exception of the

comment about repetition of the approximations to the full equation in the discussion section. I agree that the Discussion section contains a lot of summary, so I have renamed this Summary and Discussion. I believe the approximations associated with the governing equations are important and deserve to be restated in a prominent place.

Please also note the supplement to this comment:
https://www.hydrol-earth-syst-sci-discuss.net/hess-2018-242/hess-2018-242-AC2-supplement.pdf

**Supplement:**

[revised manuscript text omitted]

20scale 1D flow network systems. In the process of developing the new form, we will encounter a philosophical question as to whether the primary vertical variable in a large-scale network solution should be the depth ($H$) or the water surface elevation ($\eta$). Despite our prior work with $H$ primacy (Liu and Hodges, 2014), we shall see that there are advantages to using $\eta$ as it is identical to the piezometric pressure, which is uniform over a channel cross-sectional area. The new derivation herein provides interesting possibilities for analytically including hyperresolution bathymetric knowledge while retaining larger computational

25elements for large-scale modeling. The interaction of subgrid-scale topography with subgrid free-surface gradients is handled in a new integrated piezometric pressure term that arises in the derivation. 
[revised manuscript text omitted]

---

## Author Response (AR1)

This revision is the blue-text document that was previously uploaded as supplemental material with minor editing for grammar and spelling.

---

## Author Response (AR2)

I have made all the minor editorial corrections suggested in the final review.  Please convey my thanks to the reviewer for their careful reading and noting of these errors.

I have decided not to include the discussion of 2D models suggested by the reviewer. Firstly, I do not think it would not significantly alter the basic ideas presented for 1D models. The citations suggested are not significantly different in their methods than some of the 1D methods discussed.  Secondly, I have extensive experience in 2D modeling and I felt that to digress into it would naturally end up citing my own work and it would have the appearance of trying to increase my own citation count.

The reviewer also suggested creating some larger multi-frame figures out the the present figure set.  I generally prefer single frame figures -- however, if the editors or typesetters would prefer multi-frame I am willing to make the change.